



# Import of Atlantic Water and sea ice control the ocean environment in the northern Barents Sea

Øyvind Lundesgaard[1], Arild Sundfjord[1], Sigrid Lind[1], Frank Nilsen[2], and Angelika H.H. Renner[3]

[1]Norwegian Polar Institute, Tromsø, Norway
[2]University Center in Svalbard, Longyearbyen, Norway
[3]Institute of Marine Research, Tromsø, Norway

**Correspondence:** Øyvind Lundesgaard (oyvind.lundesgaard@npolar.no)

**Abstract.** The northern Barents Sea is a cold, seasonally ice-covered Arctic shelf sea region that has experienced major warming and sea ice loss in recent decades. Here, a two-year observational record from two ocean moorings provides new knowledge about the seasonal hydrographic variability in the region, and about the ocean exchange across its northern margin. The combined records of temperature, salinity, and currents show the advection of warmer and saltier waters of Atlantic origin into the

Barents Sea from the north. The source of these warmer water masses is the Atlantic Water boundary current that flows along the continental slope north of Svalbard. Time-varying southward inflow through cross-shelf troughs was the main driver of the seasonal cycle in ocean temperature at the moorings. Inflows were intensified in autumn and early winter, in some cases occurring below the sea ice cover and halocline water. On shorter timescales, subtidal current variability was correlated with the large-scale meridional atmospheric pressure gradient, suggesting wind-driven modulation of the inflow. The mooring records

also show that import of sea ice into the Barents Sea has a lasting impact on the upper ocean, where salinity and stratification are strongly affected by the amount of sea ice that has melted in the area. A fresh layer separated the ocean surface from the warm mid-depth waters following large sea ice imports in 2019, whereas diluted Atlantic Water was found close to the surface during episodes in autumn 2018 following a long ice-free period. Thus, the advective imports of ocean water and sea ice from surrounding areas are both key drivers of ocean variability in the region.

## 15  1  Introduction

In the Atlantic sector of the Arctic Ocean, warm and saline water masses flow in from the North Atlantic and encounter colder, fresher water formed in the polar ocean. The meeting of ocean waters of Arctic and Atlantic origin is a fundamental feature of this region. The way in which these water masses spread and interact has major impacts on the distribution of a range of ocean properties and has wider effects on ecology, biogeochemistry, sea ice, and atmospheric climate. In the relatively shallow

(∼50-450 m) Barents Sea, situated between the Arctic and the North Atlantic, most aspects of the natural environment are strongly shaped by its location at the intersection between these two ocean domains. Yet, many fundamental questions remain about the pathways of water masses in this region and the processes and dynamics that influence them — particularly in the less accessible northern region.





Atlantic Water (*AW*) is a relatively warm and saline water mass that flows into the Nordic Seas across the Iceland-Faroe-
Scotland Ridge (Hansen and Østerhus, 2000). It travels northward along the Norwegian continental slope, and from there it
enters the Arctic Ocean along two main pathways (Furevik et al., 2007; Aksenov et al., 2010, see Fig. 1 of this paper). The
first pathway is the Barents Sea Branch, by which AW travels through the southern Barents Sea. There, it spreads and cools
substantially, and mixes with local water masses before entering the Eurasian Basin mainly through the St. Anna Trough in
the Kara Sea. The second pathway is the Fram Strait Branch, which continues northwards in the West Spitsbergen Current
(*WSC*). North of Svalbard, a fraction of the WSC turns eastwards and splits in three main sub-branches near the Yermak
Plateau (Koenig et al., 2017; Crews et al., 2019). The branches crossing the Plateau ultimately converge to form the Fram Strait
Branch of the Atlantic Water Boundary Current in the Arctic Ocean (*FSAW*), which flows eastward along the continental slope
of the Nansen Basin (Pnyushkov et al., 2015; Våge et al., 2016). In contrast to the warm and saline AW are the "native" upper
ocean water masses originating in the Arctic Ocean, including its shelf seas. These water masses are commonly referred to
as Polar Water (*PW*;  Aagaard and Greisman, 1975; Rudels, 2015; Timmermans and Marshall, 2020) or Arctic Water (*ArW*;
Mosby, 1938; Lind and Ingvaldsen, 2012; Oziel et al., 2016) — we will use the former term in this study. PW is formed
through processes involving surface cooling and interactions with sea ice, meltwater, and AW, and it is generally characterized
by being cold and relatively fresh. In addition, the surface layer in seasonally ice-free areas may be subject to surface warming
in summer, resulting in warmer water masses distinguishable from AW by lower salinity.

The lateral distribution of ocean properties exerts a strong imprint on the Arctic ocean environment. The way in which water
masses are layered in the *vertical* is also of fundamental importance to how the ocean interacts with the atmosphere and sea
ice. In cold oceans, the density of seawater is dominated by salinity, and AW tends to sink below the colder but fresher PW.
This results in a subsurface temperature maximum typically found at ∼150-250 m depth in large parts of the Arctic Ocean
(Nansen, 1902; Coachman and Barnes, 1963; Timmermans and Marshall, 2020). The presence of a layer of PW atop the AW
layer prevents oceanic heat from reaching the surface, thereby reducing the ocean heat loss, sea ice melting, and warming of
the atmosphere. In the northern Barents Sea, the strength of the ocean stratification depends on the amount of sea ice that melts
in the area (Lind et al., 2018; Aaboe et al., 2021): more meltwater yields a fresher PW layer, which in turn results in stronger
stratification, limiting vertical mixing and upward heat fluxes from the AW layer. In the regions of strongest AW inflow — the
southern Barents Sea, the WSC, and occasionally the upstream regions of the FSAW — this buffering PW layer is weak or
non-existent, and AW typically extends to the surface, resulting in strong ocean-air heat fluxes with rapid cooling of the AW
and warming of the lower atmosphere (Loeng, 1991; Ingvaldsen et al., 2021).

The Barents Sea is a region where the distribution of AW and PW has particularly striking imprints on large-scale ocean, sea
ice and atmospheric conditions. The Barents Sea has two clearly distinguishable domains — Arctic in the north and Atlantic
in the south — separated by the Polar Front (Harris et al., 1998; Loeng, 1991; Oziel et al., 2016). The gradients between the
northern and southern domains are sharpest in the western part of the Barents Sea, where the polar front is largely stationary
(Harris et al., 1998). In the south, AW enters from the east through the Barents Sea Opening in large volume (∼2 Sv, Ingvaldsen
et al., 2002; Skagseth, 2008) and occupies the entire water column. Here, the AW loses heat to the atmosphere and maintains
the ocean ice-free and relatively warm through the winter (Mosby, 1962; Loeng, 1991; Smedsrud et al., 2013). In contrast, the



northern Barents Sea (*nBS*) is cold, stratified, seasonally ice covered, and dominated by PW, with increased AW influence near
the northern and southern bounds (Mosby, 1938; Loeng, 1991; Pfirman et al., 1994; Lind and Ingvaldsen, 2012). The north-south divide is also reflected in the ecological characteristics of the Barents Sea; substantial differences in food web structure and species composition largely reflect the underlying differences in the physical environment (Kortsch et al., 2015, 2019; Johannesen et al., 2017).

The Barents Sea is undergoing major, rapid ocean warming, with profound local effects on sea ice, ocean, and atmospheric
conditions (Årthun et al., 2012; Smedsrud et al., 2013; Barton et al., 2018; Lind et al., 2018). These changes in turn strongly affect ecosystem functioning and species composition in the area (Fossheim et al., 2015; Frainer et al., 2017; Dalpadado et al., 2020; Lewis et al., 2020). A fundamental change in the physical environment is the loss of sea ice in the north: despite the modest size of the region, nearly a quarter of the total Arctic sea ice area reduction in winter during the satellite era has occurred in the nBS (Docquier et al., 2020, and references therein). The loss of sea ice has strong regional atmospheric climate
effects (Ådlandsvik and Loeng, 1991; Smedsrud et al., 2013) and may affect weather in Eurasia through teleconnections (e.g. Petoukhov and Semenov, 2010; Li et al., 2018). Lind et al. (2018) documented that the ocean in the nBS has experienced rapid warming, freshwater loss, and weakened stratification after 2005, and demonstrated a link between ocean warming, freshwater loss, and decreased sea ice import. More generally, sea ice loss in the Barents Sea and along the FSAW pathway is related to increased ocean temperatures and downstream progression of warmer waters, a process known as *Atlantification* (Årthun et al.,
2012; Polyakov et al., 2017; Dörr et al., 2021).

It is well established in literature that AW enters the Barents Sea from the north as well as from the west. Hydrographic surveys from the nBS show a warming of deep (>100 m) waters towards the FSAW in the northernmost reaches of this region (Mosby, 1938; Loeng, 1991; Pfirman et al., 1994; Løyning, 2001; Lind and Ingvaldsen, 2012). While much of the northern shelf separating the FSAW from the nBS is relatively shallow, regional model simulations (Aksenov et al., 2010; Lind and
Ingvaldsen, 2012; Athanase et al., 2020) have indicated that modified AW enters through deep (>250 m) troughs that cut across the northern shelf from the continental slope into the nBS: the wide Franz-Victoria Trough and the narrower Kvitøya Trough (Fig. 1). The model results are consistent with in-situ observational evidence of AW in these troughs (Mosby, 1938; Matishov et al., 2009). Both observational (Matishov et al., 2009; Pérez-Hernández et al., 2017) and model studies (Athanase et al., 2020; Menze et al., 2020) suggest a circulation pattern within the troughs, with southward AW inflow on the western side and a return
flow of colder and fresher waters on the eastern side.

The variability of ocean transport into the nBS from the north is not well known. Aagaard et al. (1981) presented a year of point current meter data from a mooring in the centre of Kvitøya Trough, finding a weak northeastward mean flow. While this study was mainly focused on tidal dynamics, the authors noted occasional episodes of intensified northward currents during the passage of cyclones over the Barents Sea. Sternberg et al. (2001) showed a five-month record of near-bed currents measured
by a bottom tripod in Olgastretet southwest of Kong Karls Island in 1991-1992. They observed flows frequently strong enough to resuspend sediment, and frequent reversals of the near-bed currents over periods of three to eight days. Temperature records from 75 m shown in Aagaard et al. (1981) appear to indicate a seasonal cycle, with maximum temperatures observed in January, and general circulation model simulations have indicated maximum mid-depth temperatures in the nBS during autumn (Lind





and Ingvaldsen, 2012). The FSAW itself undergoes a seasonal cycle in both currents and hydrography, with stronger inflow
and a broader and warmer AW core north of Kvitøya from late autumn into early winter (Ivanov et al., 2009; Renner et al.,
2018; Lundesgaard et al., 2021). Long-term repeat hydrographic profiles collected in the nBS in autumn show substantial
temperature variability also on interannual time scales; both year-to-year variations in wind stress forcing north of Svalbard
(Lind and Ingvaldsen, 2012) and in sea ice import from the north and east (Lind et al., 2018) have been proposed as important
mechanisms for interannual variations in salinity and temperature.

A recent summary of the physical and ecological impacts of Atlantification in the Barents Sea sector of the Arctic Ocean
(Ingvaldsen et al., 2021) highlights the need for regional observational studies along advective pathways in order to better
understand the ongoing system-wide changes. The Barents Sea has been monitored extensively for half a century with annual
hydrographic surveys since the 1960s, but the current knowledge about the nBS ocean climate is largely based on ocean profiles
from the ice-free area in August-September. Ocean observations below the sea ice are sparse, and the seasonal hydrographic
variability is largely unknown. The goal of this study is to document the seasonal evolution of currents and water properties at
northern inflow regions of the nBS, focusing on the following questions:

- *How and why do ocean temperature and salinity vary through the year?*

- *What characterises ocean circulation on the northern margin of the nBS, and what forcing mechanisms drive the flow?*

- *What role do advective imports of Atlantic Water and sea ice, respectively, play in setting the seasonal hydrography of*
*the nBS?*

To this end, we present unique long-term time series of nearly continuous ocean measurements from the nBS from two
different ocean moorings deployed from autumn 2018 to autumn 2020. The moorings were located downstream of the two
deep troughs assumed to be "gateways" of AW transport to the Barents Sea from the north (based on topographic steering and
the model simulations cited above). This paper provides a detailed description of the evolution of the hydrography and currents
at the mooring sites. The findings allow us to offer an interpretation of the circulation of AW into the nBS based in the mooring
observations combined with shipboard cross-slope CTD transects. We analyse time series of sea ice concentration, sea level
pressure, and wind stress curl in order to examine drivers of the observed ocean variability, and we discuss possible mecha-
nisms by which atmospheric forcing directly or indirectly impacts the ocean in the nBS. This study aims to fill considerable
observational knowledge gaps, contributing towards a more complete understanding of the physical ocean environment of the
nBS. This is crucial for understanding ongoing processes and changes in the nBS itself, but also relevant for other Arctic shelf
seas which may come under increasing influence of Atlantic Water in coming decades.





## 2  Data and methods

### 2.1  "Northern gateway" ocean moorings

We examine the records from two ocean moorings, *M1* and *M2*, that operated in the northern Barents Sea during two consec-
utive deployments between October 2018 and September 2020 as part of the *Nansen Legacy* project's observational program.
The moorings were positioned at sites of sloping bathymetry downstream of the troughs assumed to be "gateways" of ocean
transport into the nBS from the north and east (Fig. 1). M1 (western mooring) was located at 79°35.4'N, 28°5.8'E, directly
downstream of Kvitøya Trough. M2 (eastern mooring) was situated at 79°40.7'N, 32°19.2'E, a location chosen in order to
capture inflow from Franz-Victoria Trough some 150 km to the east.

Both moorings were deployed during the KPH2018710 Nansen Legacy Paleo Cruise in October 2018 and recovered during
the KPH2019710 Nansen Legacy and A-TWAIN/SIOS-InfraNor Mooring Service Cruise in November 2019. During this same
cruise, the moorings were redeployed with a different set of instruments. Both moorings were recovered in October 2020 on
the KPH2020706 cruise led by the Institute of Marine Research. All three mooring cruises were on RV *Kronprins Haakon*. We
will refer to the individual mooring deployments as, e.g., *M1-2* (M1 mooring, second deployment).

Both moorings were equipped with one upward-looking RDI Teledyne 150 kHz Acoustic Doppler Current Profiler mounted
near the bottom, one upward-looking Nortek Signature 500 kHz ADCP mounted within the upper ∼30 m, and RBR Concerto
conductivity-temperature-pressure and RBR Solo temperature sensors at various points along the mooring line. Tables 1 and 2
show the exact locations, sensor configurations, and deployment dates of the moorings.

### 2.1.1  Moored CTD instruments

Physical variables were computed from raw Solo and Concerto data using the RBR *Ruskin* software (version 2.12.13). Sensors
were laboratory calibrated before and after deployment, and the final data were obtained by assuming a linear drift from pre-
to post-deployment calibration time series, in general yielding small corrections within the manufacturer's specified accuracy.

One Concerto sensor on M1-1 (∼170 m depth) experienced a cracked conductivity cell and showed relatively large (2.1
db) difference in pressure calculated from pre- and post-deployment calibration. Salinity from this instrument was rejected
from the analysis, and after examining the on-deck pressure record, only pre-deployment calibration coefficients were used to
compute pressure. Another Concerto sensor on M1-1 (∼250 m depth) experienced issues related to depletion of the battery
near the end of the record, and the data were rejected after 08.11.2019 as a result, shortening the record by 8 days. No data are
available from a near-bottom Concerto on M2-1 (flooded) or from Solo instruments near 60 m on both M1-2 (lost instrument)
and M2-2 (did not record).

Obvious spikes in conductivity were removed during manual inspection, and a 25-minute window median filter was applied
to all records of practical salinity. Sensor values generally compared well with shipboard conductivity/temperature/depth (CTD)
profiles collected near deployment/recovery, although the large temporal variability prevented a detailed comparison of values
below the mixed layer. For one Concerto located in the mixed layer on M2-1 (near 19 m, where water properties were relatively
stable), comparison with shipboard CTD at recovery indicated artificially low measured conductivity. We applied a drift factor





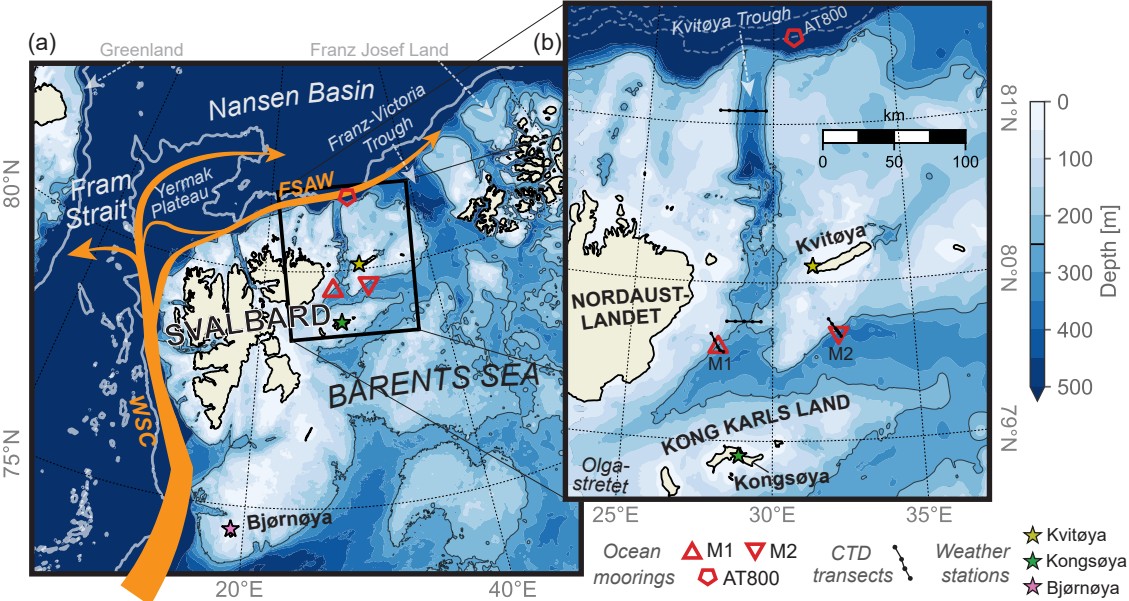

**Figure 1.** *a:* Map of the Svalbard region, indicating the West Spitsbergen Current (WSC) and its major branches entering the Arctic Ocean and recirculating in the Fram Strait (orange arrows). White and black contours show the 2000 and 250 m isobaths. *b:* Closer map of the study area in the northern Barents Sea showing positions of M1, M2, and AT800 moorings (red symbols), shipboard CTD transects (black-dotted lines) and weather stations (stars) used in this study. Bathymetry from IBCAO v3 (Jakobsson et al., 2012).

to the measured conductivity from this sensor, linearly increasing through the melt season (from 15.06.2019) to match the CTD profile at recovery on 17.11.2019, yielding a maximum change in practical salinity of 0.056 at the end of the record. Salinity variations on this scale measured by near-surface Concertos should be interpreted with caution; however, the signals discussed in this study are generally of greater magnitude.

For RBR Concertos, depth and Absolute Salinity ($S_A$), Conservative Temperature ($\Theta$), and potential density anomaly at 0 db
($\sigma_0$) were calculated using the *GSW-Python* package (https://github.com/TEOS-10/GSW-Python). For the Solo instruments, which did not have pressure sensors, instrument depths were estimated based on adjacent Concertos. The offset from planned Solo depth was obtained by interpolating between the offsets of Concertos above and, where available, below. For the bottom Solo instrument on M2-1, where no functioning Concerto was present below, an additional offset of 8.3 m was applied to the Solo depth record based on comparison with the recovery CTD profiles, reducing the deviation between median and
planned depth for this sensor to 4.1 m. These depth estimates for the Solo instruments are approximations; in many cases, the deviation between planned and observed Concerto depth was large (up to 16 m) and varied along the mooring line. We have no information about where on the mooring these depth offsets originated, and the true depth of the Solo sensors are therefore not known exactly. In this manuscript, we have used the best depth estimates described above, but also suggested a range of these estimates bounded by the depth deviations of adjacent Concertos (Table 1). Neither the visual impression of figures including





**Table 1.** Overview of moorings M1 (*a, b*) and M2 (*c, d*) with deployment dates and locations, and depths of moored CTP instruments. *Concerto*: RBR Concerto (CTP), *Solo*: RBR Solo (*T*). Median depth is calculated from pressure after removing deployment/recovery. For the Solos, depth was assigned based on interpolation between adjacent Concertos with pressure sensors (and comparison with CTD profile in the case of the bottom Solo on M2-1, see Section 2.1.1). The *Range* column shows the estimated range of the median depth of RBR Solos.

| *a)* **M1-1** - 259 m bottom depth | | | | *b)* **M1-2** - 252 m bottom depth | | |
|---|---|---|---|---|---|---|
| 05.10.2018 - 16.11.2019, 79°35.36'N / 28° 5.83'E | | | | 16.11.2019 - 21.09.2020, 79°34.98'N / 28° 4.38'E | | |
| **Instrument** | **Depth** | **Range** | | **Instrument** | **Depth** | **Range** |
| Concerto | 21 m | | | Concerto | 20 m | |
| Solo | 60 m | 55 m, 65 m | | Concerto | 97 m | |
| Concerto | 95 m | | | Solo | 155 m | 155 m, 157 m |
| Solo | 139 m | 135 m, 143 m | | Concerto | 176 m | |
| Concerto* | 170 m | | | Solo | 218 m | 215 m, 220 m |
| Solo | 210 m | 208 m, 212 m | | Concerto | 253 m | |
| Concerto□ | 250 m | | | | | |

*Suspect conductivity data; only using pressure and temperature. □The record from this instrument ends on 08.11.2019.

| *c)* **M2-1** - 357 m bottom depth | | | | *d)* **M2-2** - 360 m bottom depth | | |
|---|---|---|---|---|---|---|
| 03.10.2018 - 17.11.2019, 79°40.73'N / 32°19.19'E | | | | 17.11.2019 - 23.09.2020, 79°40.54'N / 32°18.88'E | | |
| **Instrument** | **Depth** | **Range** | | **Instrument** | **Depth** | **Range** |
| Concerto | 19 m | | | Concerto | 32 m | |
| Solo | 62 m | 50 m, 74 m | | Concerto | 111 m | |
| Concerto | 103 m | | | Solo | 174 m | 170 m, 181 m |
| Solo | 168 m | 161 m, 179 m | | Concerto | 220 m | |
| Concerto | 210 m | | | Solo | 277 m | 272 m, 282 m |
| Solo | 278 m | >270 m | | Concerto | 345 m | |

the Solo data, nor the quantitative results where they are included, are sensitive to the exact position of these sensors within the estimated depth ranges. We are therefore confident that the uncertainty in depth of the Solo instruments did not impact the results and conclusions presented in this study.

### 2.1.2 500 kHz upper ocean ADCPs

Upward-looking Nortek Signature 500 kHz 5-beam ADCPs (*Sig500s*) were mounted in the top buoys of the moorings. Data averaged from sampling over 52 second cycles every 15 minutes were used to generate a time series of near-surface ocean currents computed from the four slanted beams. Data were converted to physical variables using Nortek's *SignatureDeployment* software, and further post-processing was applied to the average profiles.





Transducer depth and bin depths were calculated from sea pressure using GSW-Python after removing a fixed atmospheric pressure. Small corrections of <1 dbar were applied to the sea pressure record from requiring zero sea pressure at recovery to

be ∼0 dbar. This nudged the offset from nominal depth toward that of nearby RBR Concertos and improved the match with the depth measured by the Sig500 centre "altimeter" beam during ice-free periods.

Measurements collected during deployment and recovery were identified from the pressure record and removed. Some individual current measurements were rejected based on a number of successive criteria for variables including backscatter amplitude, beam correlation, current speed, and instrument tilt. Measurements where bin depth was shallower than 10% of the

transducer depth were also rejected to avoid sidelobe interference near the surface. Lastly, the current vector was rotated to correct for time-dependent magnetic declination, calculated using the *geomag* python module (https://pypi.org/project/geomag/) which uses the 2015 World Magnetic Model (Chulliat et al., 2014, mean declination angles: M1-1: 19.1°, M1-2: 19.5°, M2-1: 22.2°, M2-2: 22.6°).

### 2.1.3   150 kHz water column ADCPs

RDI Teledyne 150 kHz ADCPs (*RDI150*s) were mounted near the bottom of the mooring, looking upward. All RDI150s were configured with a bin size of 8 m, obtaining current profiles from ensembles every 20 minutes (50 acoustic pings per ensemble, 24 seconds per ping). RDI150 data were converted to physical units using RDI's *WinADCP* software. Fixed depth offsets were applied to the pressure sensor-derived transducer depth and bin depths, assuming that the differences between nominal and observed depth should be the same as those of nearby Concerto pressure sensors. These depth corrections amounted to -2.3

m for both M1-1 and M1-2, and +3.2 m for M2-2. The RDI150 on M2-1 had no functioning Concerto nearby, so no fixed offset was applied to its depth record. However, a clear linear drift of 1.24 m yr −1 was found in the depth record from this instrument, which was therefore detrended assuming initially true values.

Bins with median depth located within the last 6% of the distance between the instrument and the surface were removed in order to avoid sidelobe interference (this only affected M1-1). Profiles collected during deployment and recovery were also

removed.

The RDI150 records from the first deployment were cut short by 16 days (M1-1) and 12 days (M2-1) due to battery depletion near the end of the record and associated data glitches in the period before recording ceased.

Further post-processing was applied to the RDI150 velocity data, involving a number of successive tests where measurements outside accepted ranges were rejected. Two adjoining mid-depth bins at M1-2, near 103 and 111 m depth, were suspected to be

affected by reflection of mooring elements, and therefore removed from the record. The same was the case for one near-surface bin that was removed at M1-1. A time-varying clockwise rotation of ∼20° was applied to horizontal currents from the RDI150s in order to correct for local magnetic declination, using the same procedure as described for the Sig500 data.

The flux gate compasses of the RDI150s were factory calibrated before the first deployment. Before the second deployment, the RDI150s were equipped with new batteries, but no new calibration was performed. This, along with possible compass

errors resulting from interference from magnetic elements on the buoy, prompted a closer investigation of the records of instrument heading and current direction. This investigation revealed significant deviations from expected current direction





**Table 2.** Overview of M1 and M2 mooring ADCP instruments; Nortek Signature 500 kHz ADCPs and RDI 150 kHz ADCPs. *Transducer depth* column shows median depth based on instrument pressure records. For RDI150s, depths have been adjusted to nearby RBR Concerto records as described in Section 2.1.3. *Bin depth range* column shows the median depth of the deepest and shallowest bins after processing. *Date range* column shows the time range from which data are available.

| Mooring | ADCP | Transducer depth | Bin depth range | Date range |
|---------|------|------------------|-----------------|------------|
| M1-1 | Signature 500 kHz | 20 m | 6 m to 18 m | 06.10.18 to 16.11.19 |
| | RDI 150 kHz | 247 m | 26 m to 234 m | 05.10.18 to 01.11.19 |
| M1-2 | Signature 500 kHz | 19 m | 4 m to 16 m | 17.11.19 to 21.09.20 |
| | RDI 150 kHz | 251 m | 23 m to 239 m | 17.11.19 to 21.09.20 |
| M2-1 | Signature 500 kHz | 19 m | 4 m to 16 m | 04.10.18 to 17.11.19 |
| | RDI 150 kHz | 344 m | 116 m to 324 m | 03.10.18 to 05.11.19 |
| M2-2 | Signature 500 kHz | 31 m | 5 m to 29 m | 18.11.19 to 23.09.20 |
| | RDI 150 kHz | 349 m | 121 m to 337 m | 17.11.19 to 23.09.20 |

based on bathymetry, upper ocean currents measured by the Sig500s, and the Arc5km2018 tidal model (Erofeeva and Egbert, 2020). In particular, mean and tidal currents measured at M2-2 were oriented ∼70° clockwise of the expected direction and of the direction measured during M2-1. Compass direction issues are a frequent concern for ADCP current measurements (which are sensitive to compass heading) and pose an especially major challenge at high latitudes where the horizontal magnitude
of the Earth's magnetic field is weak. As a corrective measure, we adjusted RDI150 current directions based on the observed relationship between the currents measured in the upper bins of the RDI150s and those measured by the overlying Sig500s, which were equipped with solid state magnetometer compasses and exhibited more consistent current records. The method was based on von Appen (2015), who applied a heading-dependent correction based on a best-fit relationship between current
directions measured by moored ADCPs and fixed-point current meters on the same moorings. We compared current directions from the upper RDI150 bins with those from the Sig500s, extracting a relationship between the two as a function of the heading of the RDI150 instrument. Only data from periods of near-full ice cover (and weak vertical gradients) were included in the comparison. A best fit to the direction difference between the RDI150 and Sig500 currents was extracted using non-linear least squares weighted by current speed, assuming the functional shape of a single period sinusoid with an offset. The extracted
heading-dependent current direction offset was then applied to the entire record of horizontal currents from the RDI150. This procedure visibly improved the records at the M1 mooring, where the RDI150 and Sig500 ranges were separated by <10 m and the heading-dependent direction difference exhibited a clear sinusoidal shape. At M2, the RDI150 and Sig500 ranges were separated by >100 m and the relationship between the two instrument records was noisier. Yet, the correction clearly improved the match between M2-2 mean currents and the local bathymetric slope, between M2-2 tidal currents and
Arc5km2018 predicted tides, and between M2-1 and M2-2 currents overall.





Although we are confident that these corrective measures markedly improved the RDI150 current directions, uncertainties are large in the case of M2. While we have no procedure to rigorously estimate true errors, we believe the direction of RDI150 currents measured at M1 to be a good representation of the true flow direction. The error is greater for M2-1, which had a recent calibration but could only be compared directly with currents further up in the water column. The greatest uncertainties are

associated with M2-2, where we lack a compass calibration after the battery change and where we haev applied relatively large corrections to the compass data. The RDI150 records from M2 should therefore only be used to indicate the current direction in a very general sense.

### 2.1.4  Interpolation and filtering

All variables were linearly interpolated onto equal grids in time and depth for analysis and plotting. All currents shown in this

manuscript have been filtered in time using a 40-hr 7th order Butterworth filter before further analysis in order to focus on the subtidal component of the flow.

### 2.2  Slope current mooring

In order to compare the seasonal evolution in the nBS with the AW slope current north of Svalbard, we used ocean current and temperature data from the *AT800* mooring, which was located on the slope north of Kvitøya at 81°32.9'N, 31°52.3' E as part

of the *A-TWAIN* mooring array (Renner et al., 2018), which has been in place with a varying number of operational moorings since 2012. The AT800 mooring was located at a water depth of approximately 880 m, near the core of the FSAW (Fig. 1). Here, we use the depth-average temperature and currents between 125 m and 230 m as supporting information about the AW circulation northeast of Svalbard during the study period to aid the interpretation of the records from M1 and M2.

The AT800 mooring was deployed and recovered during the same cruise as M1-2 and M2-2, and data are available from 21

Nov 2019 to 29 Sep 2020. We use current data from an upward-looking RDI Teledyne 150 kHz ADCP mounted near 297 m depth. The ADCP collected current profiles in 8 m vertical bins every 20 minutes (37 pings per ensemble, 32.4 seconds per ping). Temperature data were obtained from four RBR Concertos located at median depths 30 m, 138 m, 190 m, and 292 m, collecting one sample every 5 minutes. Currents and temperatures from the AT800 mooring were processed and interpolated in the same way as for the gateway moorings. A small (<10°) heading-dependent compass correction was applied to ADCP

current direction based on on-shore, post-deployment calibration against an external compass before removing the batteries.

### 2.3  Shipboard CTD transects in November 2019

Cross-slope transects of shipboard CTD profiles were conducted at the M1 and M2 sites during the KPH2019710 cruise in November 2019. In addition, two zonal transects were performed across the southern (79°45' N) and northern (81°05' N) parts of Kvitøya Trough (Fig. 1b) during the same cruise. CTD variables were processed and calibrated against in situ salinity

samples by the Institute of Marine Research. Profiles were conducted through the moon pool of the ship; the upper 15 m of each profile were therefore discarded.



## 2.4 Sea ice and atmosphere

The dynamics and hydrography of the Arctic Ocean are strongly influenced by interactions with the sea ice cover and the atmosphere. We therefore examined datasets capturing the sea ice state and atmospheric variability in the study region to provide context for the ocean observations. We also evaluated the covariability of specific time series from these datasets in order to examine potential driving factors of ocean variability.

The evolution of the large-scale sea ice cover was assessed from AMSR2 sea ice concentration based on the ASI algorithm, version 5.4 (*AMSR2 SIC,* Spreen et al., 2008). Data are available at daily resolution with a spatial resolution of 6.25 km. Gridded data were bilinearly interpolated to the mooring locations to obtain local SIC time series. A regional index of sea ice concentration in the northern Barents Sea was obtained by averaging over all non-land grid points between 78° N to 80° N and 25° E to 35° E (*A-nBS* area, Fig. 2d-h). In addition, an index of the sea ice north of Svalbard was obtained by averaging over all non-land grid points between 80.0° N to 81.5° N and 12° E to 29° E (*A-nSVB* area, Fig. 2d-h).

We examined wind, pressure, and temperature data from three automatic weather stations (*AWS*s) operated by the Norwegian Meteorological Institute, located on Kvitøya (80°6' N, 32°27' E), Kongsøya (78°54' N, 28°53' E) and Bjørnøya (74°30' N, 19°00' E). Weather station data were downloaded from eKlima.no and converted to daily means for analysis.

Regional sea level pressure and 10-m winds were obtained from the C3S Arctic Regional Reanalysis product (*CARRA,* Copernicus Climate Change Service, 2021), whose *East* domain encompasses the study region. Wind stress was computed from daily averaged 10-m wind components (supplied at 2.5 km resolution) using the parameterization of Large and Pond (1981), and wind stress curl was calculated from the wind stress components. Since the atmospheric pressure from the weather stations at Bjørnøya and Kvitøya will be compared with wind stress curl from CARRA, we confirmed that the daily pressure record from CARRA matched the observations from the weather stations ($r > 0.99$ for both weather stations, 2018-2020).

## 2.5 Correlations

By *correlation*, we will in this paper imply a value of Pearson's correlation coefficient, $r$. Correlations were considered *statistically significant* if a two-sided Student-t test yielded $p < 0.05$ for a null hypothesis of zero correlation. Since successive daily measurements of atmospheric and oceanic variables are not typically truly independent, tests were conducted using an *effective* sample size calculated as the total sample size divided by an integral time scale found by summing the product of the autocorrelation coefficients of the two time series (Davis, 1976) up to the lag where this product is no longer of positive sign.

# 3 Results

## 3.1 Regional sea ice and atmospheric conditions 2018-2020

Ocean circulation and water mass formation in the polar oceans are influenced by interactions with the atmosphere and sea ice cover (e.g., Timmermans and Marshall, 2020). We therefore give a brief description of the regional atmospheric and sea ice conditions during 2018-2020 to establish the environmental context of the ocean observations. The evolution of sea ice

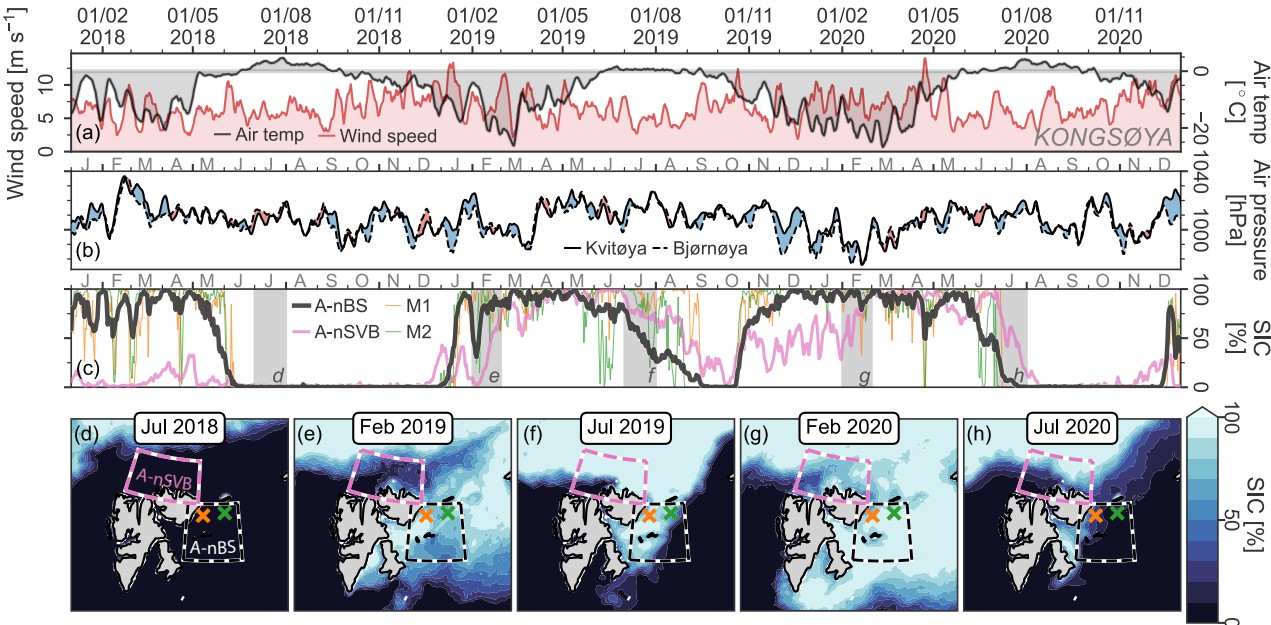

**Figure 2.** Atmospheric and sea ice conditions 2018-2020. *a:* 7-day averaged air temperature and wind speed from the Kongsøya AWS. *b:* 7-day averaged air pressure from the Kvitøya and Bjørnøya AWSs, with the difference between the two indicated with colors: blue (red) indicates higher (lower) pressure at Kvitøya. *c:* Time series of daily AMSR2 sea ice concentration: *Black:* average SIC in the A-nBS area in the northern Barents Sea (black dashed polygon in d-h). *Pink*: Average SIC in the A-nSVB area north of Svalbard (pink dashed polygon in d-h). Thin lines indicate SIC interpolated to the M1 (orange) and M2 (green) coordinates. *d-h:* Maps of monthly average July and February AMSR2 SIC in the broader area from 07/2018 to 07/2020. The time corresponding to each map is shown as gray shading in c.

concentration in the A-nBS area, encompassing the moorings, and the A-nSVB area north of Svalbard, upstream along the AW current (Fig. 2c), show the large year-to year differences in regional sea ice extent. In 2018 the A-nSVB area remained

largely ice-free the entire year. The A-nBS area was largely ice-covered (50% to 100% SIC) in winter and spring of 2018, but experienced a long ice- free season lasting from June 2018 into January 2019 (Fig. 2c). In the following season, both areas were largely ice-covered from March to July 2019. During all of 2019, the A-nBS area was completely ice-free only during a short period in September-October. The A-nSVB area remained partially open from autumn 2019 until February 2020, with mean SIC fluctuating around ~50% (Fig. 2cg). During both 2019 and 2020, the A-nBS area became sea ice free approximately

one month prior to A-nSVB.

   Weekly mean air temperatures in the nBS, measured by the Kongsøya AWS, reached minima of <-20°C in March/April during all three years and peaked with temperatures exceeding +4°C in August in 2018 and 2020 (Fig. 2a). Summer air temperatures were relatively low in the high sea ice year of 2019, remaining below +1°C with one single exception in June. Wind speeds measured at the same weather station generally increased in strength and variability during autumn/winter (Fig.



2a), with monthly average speeds ranging from 4 m s$^{-1}$ to 10 m s$^{-1}$. The seasonal pattern is consistent with the increasing impact of storm systems during autumn and winter (Wickström et al., 2020).

In Section 3.4, we detail a correlation between ocean currents and the difference in atmospheric pressure between Kvitøya and Bjørnøya. To analyse whether the regional meridional atmospheric pressure gradient has an impact on the inflow of AW to the Barents Sea from the north, we defined a quantity $\Delta p_{K-B}$: the daily mean atmospheric pressure at the Kvitøya AWS

minus the daily mean atmospheric pressure at the Bjørnøya AWS. Positive (negative) values are shown in blue (red) in Fig. 2b. Atmospheric pressure at the two locations generally varied in concert, but there were also periods where they diverged, typically as a result of the passing of synoptic-scale weather systems. In particular, positive values of $\Delta p_{K-B}$ were often associated with the passage of low-pressure systems from the west into the Barents Sea (see Wickström et al., 2020; Madonna et al., 2020).

The meridional gradient in surface atmospheric pressure captured by the $\Delta p_{K-B}$ index is closely related to the wind in the nBS. The zonal component of the winds captured by the Kongsøya AWS was strongly and significantly correlated ($r = 0.74$) with a positive $\Delta p_{K-B}$; i.e., lower pressure in the southern parts of the Barents Sea is associated with easterly winds in the nBS, consistent with geostrophic air flow.

## 3.2 Mooring records of water masses and ocean currents

### 3.2.1 Seasonal evolution of currents and temperature at M1

We begin with describing the record from the M1 mooring, where the signature of Atlantic Water inflow was the strongest. The overall evolution of sea ice concentration, ocean currents, and water temperature and salinity at M1 is shown in Fig. 3. There is an apparent relationship between the seasonal evolution of currents and temperature, with southwestward flow generally corresponding to warmer mid-depth waters. Temperature and currents can also be seen to vary in concert on shorter timescales

(days to months) at various points in the record.

Ocean currents (Fig. 3c) were largely uniform throughout the water column through most of the study period. For instance, the daily mean subtidal currents along the direction of maximum subtidal variance at 115 m depth were strongly correlated with the currents at both 15 m ($r = 0.68$) and 215 m ($r = 0.77$). The predominant direction of the subtidal currents was towards the southwest, with the maximum variance of the currents averaged over the RDI150 depth range oriented along the axis 45.2°

CCW of east-west. This direction aligns approximately with the local 250 m isobath, which runs parallel to the southeastern coast of Nordaustlandet (Fig. 1a), suggesting strong topographic steering of the flow.

To a first approximation, currents and temperatures were seasonally modulated, with the strongest southwestward flow in autumn/winter coinciding with higher ocean temperatures. The weakest flow in spring/early summer was associated with a colder and more homogenous water column. However, there was both considerable variability on shorter timescales and

large interannual differences within the study period. The strongest southwestward flow occurred in late 2018: the current (specifically: the depth-mean daily subtidal current along the primary axis, positive towards the northeast) was on average -10.0 cm s$^{-1}$ from October 5 through December 31, with a southwestward maximum of -25.1 cm s$^{-1}$ on November 12. This





period of strong southwestward flow coincided with the warmest water column in the record, with in-situ temperatures >2.5°C between 100 and 200 m depth during much of October 2018. From November 1 to December 31, ocean temperatures gradually

decreased, but remained above -0.54°C through the entire water column. In addition to this slow decrease in temperature, shorter periods of reduced or reversed flow were associated with low temperature anomalies and vice versa — a tendency that also occurs later in the record (compare panels c and d in Fig. 3).

The warm period in late 2018 was followed by an abrupt reversal of the flow in January 2019, reaching a northeastward maximum of 13.8 cm s$^{-1}$ on January 12. The northeastward currents subsided within ∼10 days, and the flow remained very

weak in the following five months (2019 Feb-Jun mean: -1.7 cm s$^{-1}$). The abrupt flow reversal in January coincided with *i)* the arrival of sea ice, and *ii)* intensified cooling of the upper ocean. Meanwhile, the ocean below 100 m depth continued to cool, but more gradually. Some water warmer than -1°C remained in this deeper layer through February and March, but between April 5 and August 1, ocean temperatures were generally below -1.5°C throughout the water column, with some minor exceptions near 100 m depth. During this cold period, temperatures were stable compared to the more variable autumn and early winter

temperatures.

Currents increased in strength again from mid-July 2019, and southwestward flow dominated through the short ice-free period of September-October 2019, and into November. A mid-depth temperature maximum reappeared after the onset of the southwestward flow, but temperatures remained below 0°C until the turnover of the mooring in mid-November. After this, the currents became more variable, including occasional flow reversals. Unlike the preceding year, the southwestward flow

persisted through most of winter and spring 2020. Mid-depth temperatures were substantially higher after the redeployment of the mooring in November 2019, with maxima of ∼2±1°C through January 2020. Ocean temperatures remained elevated in winter and spring 2020 compared to the corresponding period in 2019. A deep, relatively warm (>1°C) layer persisted until mid-March, and, with a few exceptions, maximum temperatures below 150 m depth remained above -0.5°C —far warmer than during the preceding spring—until the recovery of the mooring in September 2020. The southwestward flow ceased in early

May 2020 after a period of intensified currents (up to -20 cm s$^{-1}$) in late April. This short episode of increased southwestward flow coincided with a temporary decrease in sea ice concentration and a brief increase in temperatures throughout the water column.

Very weak northeastward subtidal currents dominated from May until the opening of the sea ice in July (May-July 2020 mean: +0.9 cm s$^{-1}$), a period in which a thick (>100 m), cold (<-1.7°C) winter mixed layer overlaid the warmer deep water.

Thereafter, the southwestward flow gradually intensified, accompanied by a moderate warming of the water column, up until the recovery of the mooring in late September. Interestingly, the period of July to September 2020 was the only part of the M1 record where we observed clear evidence of surface warming. As a result, the water column during this period exhibited a mid-depth temperature minimum typical of the remaining "winter water" often found in polar regions, including the nBS, during late summer.



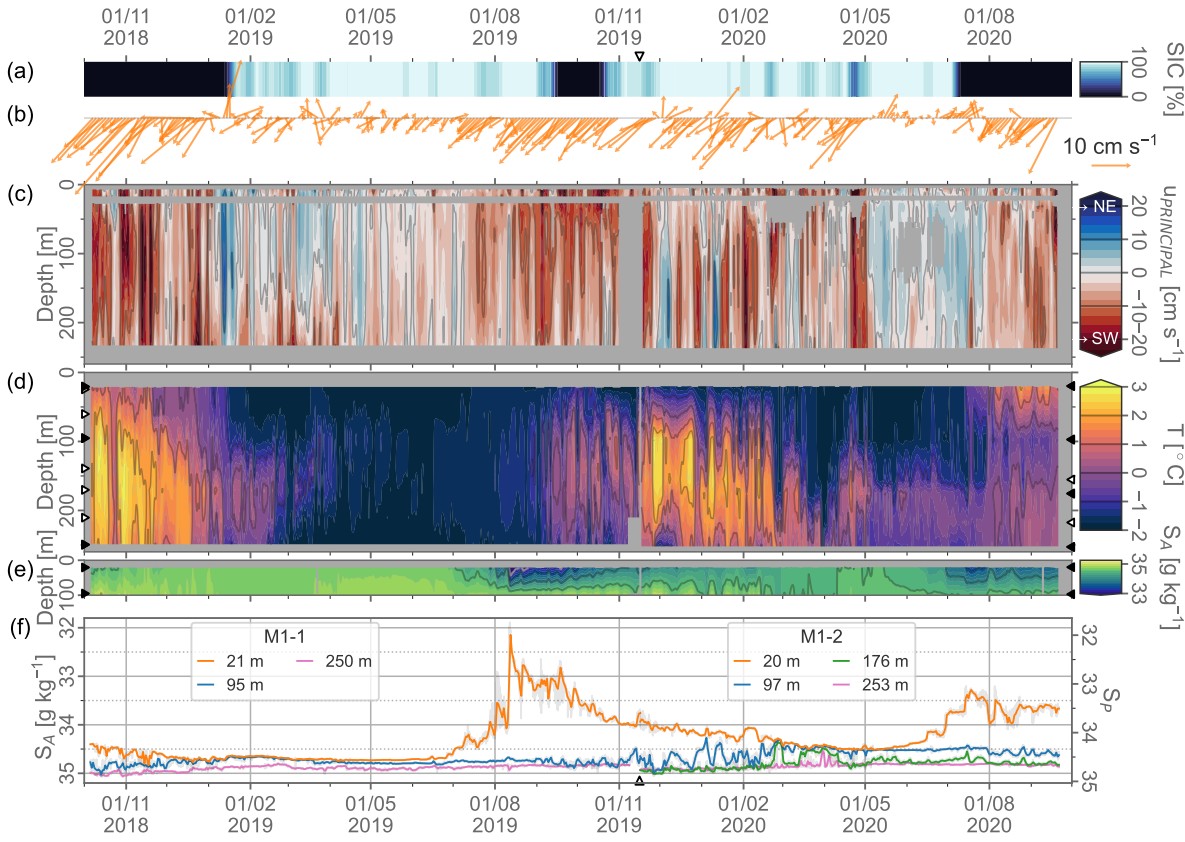

**Figure 3.** 7-day averaged time series from the M1 mooring. *a:* Sea ice concentration at the mooring site from UoB AMSR2, *b:* Depth averaged current vector from the RDI150 instrument (29 m to 231 m depth), *c:* currents along the direction of maximum subtidal variance (positive/blue 45.2° CCW of E), *d:* in-situ temperature, *e:* contours of Absolute Salinity in the upper 100 m, and *f:* Absolute Salinity measured at each individual sensor. In d) and e), mean position of RBR Concerto instruments during the two deployments are indicated with black triangles on either side. The position of RBR Solo instruments (whose vertical position is less certain) is indicated with white triangles.

### 3.2.2 Salinity and upper layer freshening at M1


The salinity records from M1 show strong interannual variability over the study period (Fig. 3ef). Particularly striking is the upper layer, which was significantly fresher after the melt season of 2019 than before. The near-surface layer was relatively fresh at the start of the record, with $S_A$ of 34.4 g kg$^{-1}$ measured by the sensor at 21 m depth. Salinity was highly variable in autumn 2018 (see Section 3.2.5) but generally increased before stabilizing near 34.7 g kg$^{-1}$ from December until the following

summer. The near-surface waters then began to freshen in the end of June 2019, and $S_A$ dropped substantially in the following months, reaching a minimum of <31.9 g kg$^{-1}$ on August 11, before the disappearance of sea ice in early September. $S_A$



near 20 m depth then increased gradually over the following 7 months, reaching a maximum of ~34.5 g kg$^{-1}$ in April 2020. Upper-ocean $S_A$ began to slowly decrease again in late May, coinciding with the sea ice melting, before dropping rapidly to a minimum of <33.2 g kg$^{-1}$ in July and August. The near-surface water remained relatively fresh ($S_A$ <34.0 g kg$^{-1}$ with a few

brief exceptions) until recovery of the mooring in late September 2020.

Deeper in the water column, the record begins with a period of high salinity, with global maxima of $S_A$ (>35.05 g kg$^{-1}$) occurring near 250 m depth in October 2018. These high salinities coincided with the higher temperatures and increased southwestward flow observed during this period (Fig. s 3c, d). This relationship between positive anomalies of $S_A$, temperature, and southwestward currents is also present in other parts of the record below 100 m (compare panels c, d, and f of Fig. 3c,

d). $S_A$ near 250 m decreased gradually between November 2018 and February 2019, before increasing from ~34.8 g kg$^{-1}$ to ~34.9 g kg$^{-1}$ in early March and remaining on this level through June 2019. Mid-depth salinity increased slightly along with the reappearance of a mid-depth temperature maximum in autumn 2019. During November 2019-January 2020, mid-depth salinity was highly variable, and generally covaried with temperature. The highest salinities during the second deployment ($S_A$ ~35.0 g kg$^{-1}$ at 175 m depth) were observed during the temperature peak in November 2019. During spring 2020, mid-

depth salinity stabilised at a lower level (~34.5 g kg$^{-1}$) than during the preceding year (~34.8 g kg$^{-1}$), with some exceptions during the warm anomaly in April 2020.

The difference in salinity between the years in the record is illustrated by estimating the freshwater content (*FWC*) with the formula:

$$FWC = \int_{z0}^{z1} \frac{S_{A,\,ref} - S_A}{S_{A,\,ref}} \ dz \tag{1}$$

Assuming a linear profile between daily mean salinities at the three sensors capturing salinity, integrating between 25 m and 245 m, and using a reference salinity $S_{A,\,ref}$ = 35.05 g kg$^{-1}$, this crude calculation suggests that the mean freshwater content in the water column below 25 m increased by 1.3 m from Oct 7 - Nov 7 2018 to Oct 7 - Nov 7 2019; more than a doubling from the total FWC in Oct 7 - Nov 7 2018 (1.2 m).

### 3.2.3    Comparing M1 and M2

At the M2 mooring, 85 km further east in the nBS, the evolution of currents and water masses showed many similarities to that at M1. At both locations, the predominant current direction was towards the southwest (Fig. 4bc). The axis of maximum current variance within the RDI150 depth range at M2 was oriented along the axis 35.7° counterclockwise of east-west, with the mean currents again corresponding approximately to cyclonic flow along the local bathymetric contours. The current records from M1 and M2 are not entirely comparable (the range of the RDI150 ADCP at M2 only extended up to ~120 m), but it is clear that

the depth-average currents at M2 were also strongest in autumn/winter 2018, when a persistent southwestward flow coincided with warmer ocean temperatures (Fig. 4cd). After this initial period, currents at M1 were generally stronger and more variable on all subtidal timescales. From January 2019, the current record from M2 is dominated by a weak (5-10 cm s$^{-1}$) flow at depth,

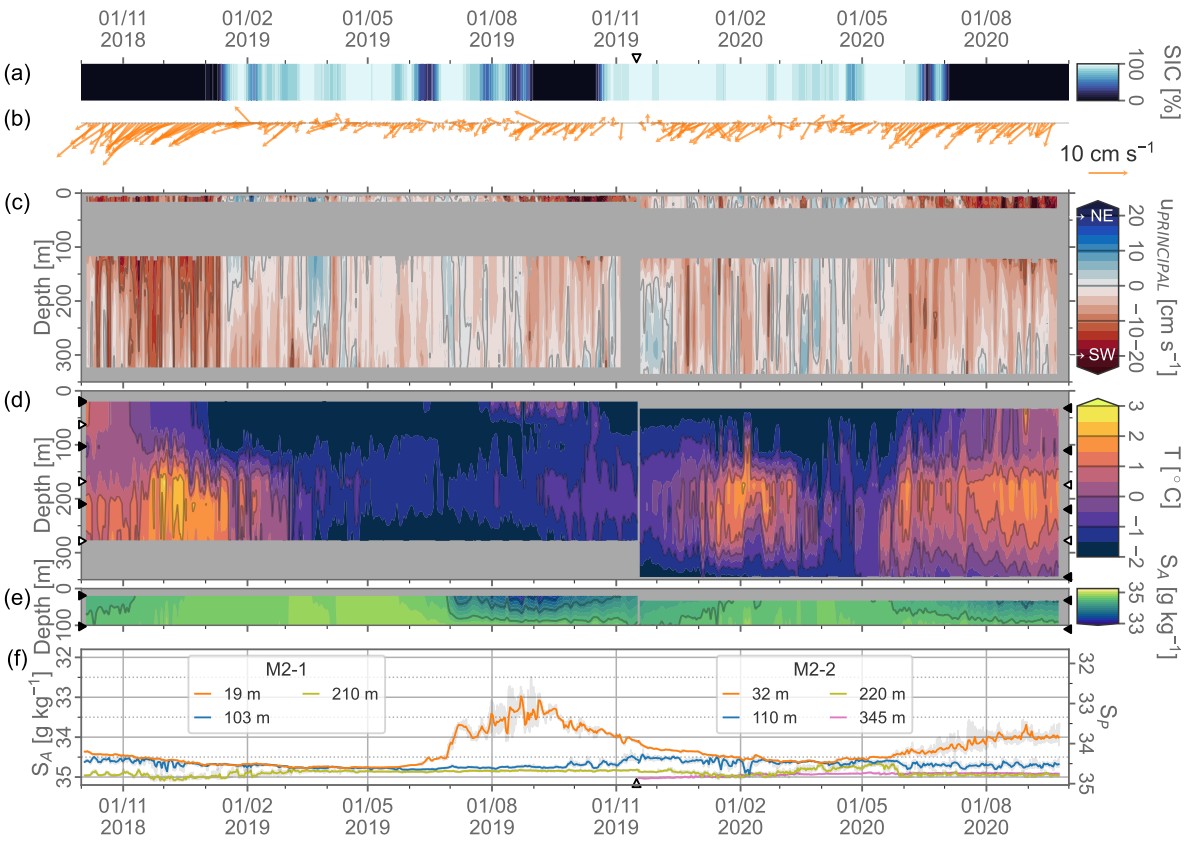

**Figure 4.** 7-day averaged time series from the M2 mooring. Like Fig. 3, but with a greater y-scale, and the depth average current in b) is calculated between 123 m and 321 m. The direction of maximum subtidal variance (positive/blue in panel c) is 35.7° CCW of E. Note that the uncertainty of the >100 m current direction during the second deployment of M2 is large.

and it shows neither the extended periods of strong southwestward flow intensification nor the periods of cessation or reversal observed at M1. The near-surface currents at M2, measured by the upper ocean ADCP, show a much stronger seasonal cycle,
with ice-free periods in autumn corresponding to intensified flow towards the southwest.

The records from M2, like those from M1, show a mid-depth temperature maximum that weakened and strengthed several times over the study period. Temperature maxima were found somewhat deeper at M2 ($\sim$200 m) than at M1 ($\sim$150 m). The mid-depth temperature maximum at M1 frequently extended into the upper 50 m during warm periods, while warmer waters at M2 were largely restricted to >150 m depth, underlying a cold, uniform layer for most of the study period. A notable difference
between the two sites was observed in summer/autumn 2020; a clear temperature maximum ($T$ >1°C) was present at depth at M2 through most of June-September, while temperatures largely remained below 0°C at M1. Temperatures also remained

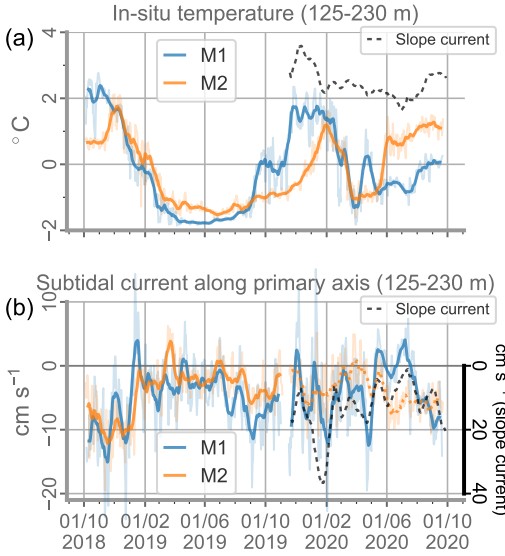

**Figure 5.** Time series from the M1, M2 and AT800 (slope current) moorings of *a)* in-situ temperature and *b)* subtidal currents along the principal axis, both depth-averaged between 125 m and 230 m after linear interpolation between available sensors. Thin, transparent lines show daily mean quantities, thick lines show a 15-day running mean (currents from M2-2 are shown as a dotted line to indicate the substantial uncertainty). Note the separate scale for the AT800 slope current in panel b.

slightly higher at M2 than M1 during spring/summer 2019; a weak mid-depth temperature maximum ($T$ >-1.5°C) remained at M2 during this time. Like at M1, mid-depth salinity at M2 covaried with temperature, with the highest salinities ($S_{A}$ >35 g kg$^{-1}$ near 200 m depth) occurring during periods of high mid-depth temperature.

Despite the differences outlined above, the overall timing of fluctuations in currents and temperature were similar at the two moorings, although both amplitudes and variability were higher at M1. Figure 5a shows that while mid-depth temperatures were higher at M1 in autumn 2018, temperatures decreased at similar rates at both moorings through winter 2019. Currents (Fig. 5b) also evolved similarly at the two sites in the first year of the joint record, but sub-monthly variability was more pronounced at M1. In autumn 2019, mid-depth warming began earlier and more abruptly at M1, with a sudden temperature

increase of ∼1.5°C after a period of intensified southward flow, and a second increase of the same magnitude in November, a period of high-amplitude flow variability. During this time, currents remained steadier at M2, and the temperature only increased gradually until the warming rate picked up during December-January. M2 mid-depth temperatures reached a peaked maximum ($T$ >1.5°C) around February 1 2020, after which the temperatures at the two locations again decreased more or less in unison. From April 2020 onward, the temperature and current records diverge between the two moorings, with periods of

elevated southwestward flow and increased temperature occurring first at M1 (April-May) and then at M2 (June-September).

Also shown in Fig. 5 are temperatures and currents from the AT800 mooring on the shelf slope. The principal axis direction at the AT800 was positive for current direction 49.9° CCW of east. The AT800 mooring was located in the FSAW slope current,





and generally showed higher temperatures and higher flow speeds than both nBS moorings. An episode of intensified flow in January 2020 had no obvious counterpart at M1 or M2, although the southwestward flow at M1 was relatively strong in this

period. Temperature increases at AT800 in autumn 2019 and late summer 2020 coincided approximately with warming periods at M1.

Upper ocean salinity evolved similarly at the two nBS moorings; strong summer freshening of the near-surface layer occurred at nearly the same time at M1 and M2 both in 2019 and 2020. The clearest difference between the salinity records is that mid-water salinity was lower at M1 than M2 during spring 2020 (compare Figures 3e and 4e). The apparent downward mixing of

fresher water at M1 during this period was not observed at M2. Salinity at the uppermost sensor at M2 remained around 34.5 g kg$^{-1}$ from February until the subsequent melt season — lower salinity at M1 may indicate that the upper ocean was fresher at M1 than at M2 at this time, but since the sensors were located at depths separated by ~10 m, a direct comparison is not appropriate.

### 3.2.4    Water mass distribution

The distribution of $\Theta$ and $S_A$ at the mid-depth sensors is shown in Fig. 6. Since the sensor depths varied, these distributions cannot be compared directly between moorings and deployments; however, they do provide useful insight into the seasonal water mass variations at mid-depth at the mooring sites.

The distributions are shown along with the water mass classification system for the broader Barents Sea and Nansen Basin region suggested by Sundfjord et al. (2020). We interpret observed water mass distribution in Fig. 6 primarily in terms of

two water masses: *Atlantic Water* (AW, $\Theta$ >2°C, $S_A$ ≥35.06 g kg$^{-1}$), which originates at lower latitudes and exists on the continental slope north of Svalbard as well as south of the Polar Front, and *Polar Water* (PW, $\Theta$ ≤0°C, $\sigma_0$ ≤27.97 kg m$^{-3}$), which is formed in the Arctic Ocean including its shelf seas through cooling and interactions with sea ice and meltwater.

In general, the mid-water $\Theta$-$S_A$ distributions at M1 and M2 can be described as a time-varying mixing between these two water masses. However, AW was always modified and almost never present in undiluted form following the strict definitions

given above. Throughout the study period, mid-water properties at the moorings largely fell along a trajectory extending from the surface freezing point in the 34.5 g kg$^{-1}$ to 35.0 g kg$^{-1}$ $S_A$ range, curving upwards and rightwards in $\Theta$-$S_A$ space towards the AW classification. Broadly speaking, the transition occurred along isopycnals in the vicinity of $\sigma_0$ =27.8 kg m$^{-3}$ (with several exceptions detailed below). Along much of this trajectory, water masses fell within the nominal bounds of *warm Polar Water* (*wPW*), which can also originate from solar heating of PW. However, as these movements in $\Theta$-$S_A$-space were largely

isopycnal, we interpret them as a signature of mid-water mixing between PW and AW rather than as a result of surface heating — consistent with the measurements being conducted well below the direct influence of surface forcing.

In autumn 2018, the AW influence at M1 was relatively large, with a $\Theta$-$S_A$ distribution extending all the way into the AW range (Fig. 6a). Through the following winter, the water cooled and freshened until the distribution became largely clustered near the freezing line. At the bottom sensor (near 250 m), salinity increased, and in May 2019, $\Theta$-$S_A$ properties frequently fell

within the range defining *Cold Barents Sea Deep Water* (*CBSDW*, $\Theta$ ≤-1.1°C, $\sigma_0$>27.97 kg m$^{-3}$), indicating the influence of brine rejection and/or mixing with interior deep waters of the nBS. The following periods (Fig. 6cd) show the reappearance of


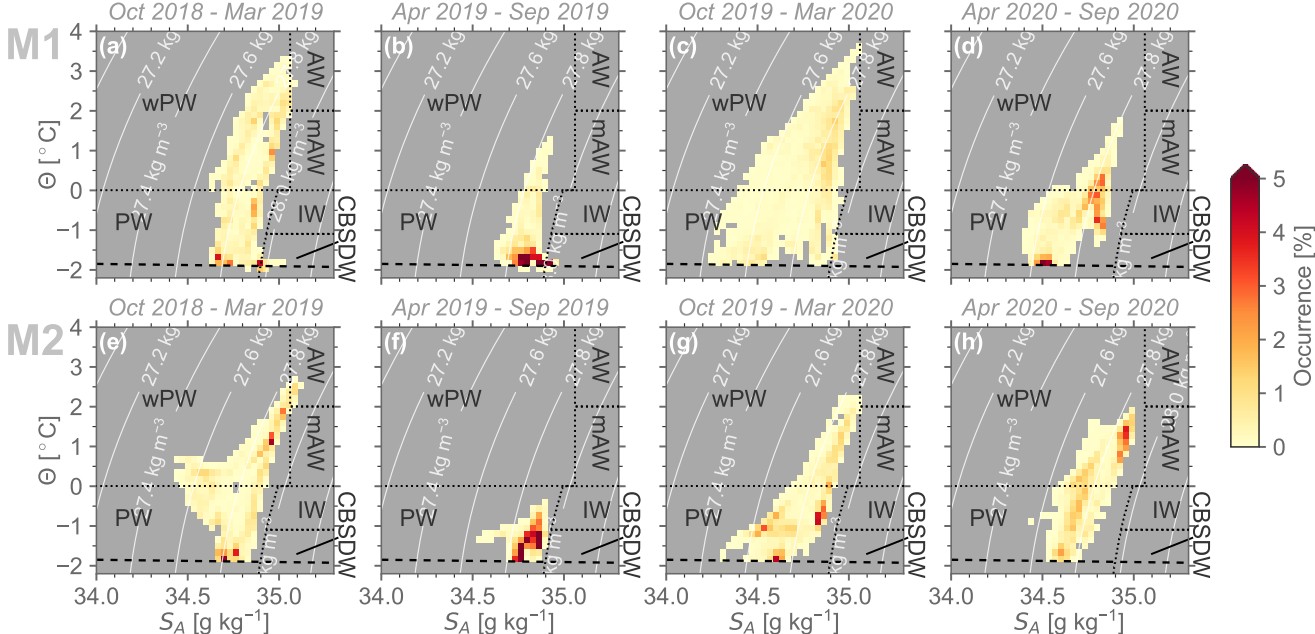

**Figure 6.** Distributions of Absolute Salinity ($S_A$) and Conservative Temperature ($\Theta$) in hourly mean data from moored RBR Concerto sensors on the M1 (upper panels) and M2 (lower panels) moorings. Columns show separate 6-month intervals (see panel titles). Only including sensors between 90 and 260 m depth (number and depths of sensors vary, see Table 1). Water mass categories from Sundfjord et al. (2020): *AW:* Atlantic Water, *mAW:* Modified Atlantic Water, *PW:* Polar water, *wPW:* Warm Polar Water, *IW*: Intermediate Water, *CBSDW:* Cold Barents Sea Dense Water. Dashed black line shows the surface freezing point.

a weaker AW signal in autumn 2019, followed by a stronger AW signal as well as a greater spread towards lower salinities as a result of the higher amount of freshwater in the water column (Fig. 3e).

A similar temporal evolution occurred in the $\Theta$-$S_A$ distribution at M2, with some notable differences. In the beginning of
the record, relatively fresh ($\sim$34.6 g kg$^{-1}$) and warm ($\sim$0.25°C) water was present near 100 m depth, resulting in a protrusion of the distribution into the fresher wPW range, likely indicating a degree of surface influence (Fig. 6e). The AW core at M2 was also of higher salinity than at M1 during the first months of the record.

The weak mid-depth warming observed at M1 in early autumn 2019 was nearly absent at M2, resulting in a more restricted distribution, clustering near $S_A \sim$ 34.8 g kg$^{-1}$ , $\Theta \sim$ -1.5°C (Fig. 6f). The freshening of the water column at M1 in winter
2020 was less pronounced at M2, and for the remaining duration of the study period, the distribution at M2 fell largely along the PW-AW line (Fig. 6gh).

The $\Theta$-$S_A$ distributions from the shallowest sensors, near 20-30 m depth (Fig. 7), reflect the wider salinity range in the upper ocean. During the mostly ice-covered period of April 2019 through March 2020, distributions in the upper ocean were generally clustered close to the freezing line and within the PW water mass boundaries. Some exceptions occurred at M2 due to surface



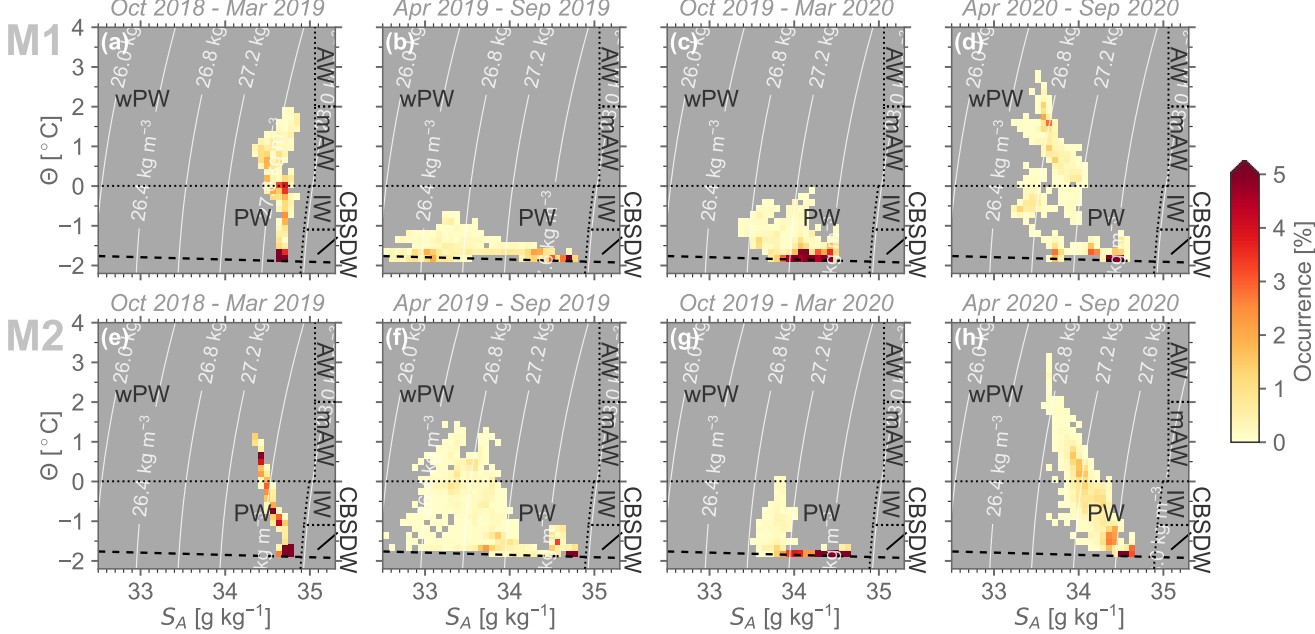

**Figure 7.** Distributions of Absolute Salinity ($S_A$) and Conservative Temperature ($\Theta$) in hourly mean data from moored RBR Concerto sensors on the M1 (upper) and M2 (lower) moorings, only including data from the uppermost Concerto sensors (median depths between 19 and 30 m). See Fig. 6 for details, but note the different x-axis scale.

warming in ice-free periods in summer, which resulted in excursions into the wPW range at $S_A$ <34 g kg$^{-1}$. During autumn 2018 and summer 2020, temperature ranges were comparatively wide, and salinity ranges comparatively narrow. Warming of the upper ocean generally occurred along with freshening, resulting in distributions curving leftward and upward. A notable exception found at M1 during autumn 2018, warrants additional attention:

### 3.2.5 Episodes of Atlantic Water influence in the upper ocean at M1 during autumn 2018

In general, increases in temperature observed at the uppermost sensors were associated with decreases in salinity (Fig. 7). An exception to this tendency was found in autumn 2018 at M1, where abrupt increases in temperature of up to >1.5°C coincided with increases in salinity of up to >0.3 g kg$^{-1}$(Fig. 8). Episodes of warming and increased salinity occurred particularly in late October and November. The two major warming episodes captured by the upper ocean sensor at this time occurred during periods of increased southwestward flow throughout the water column (Fig. s 3c, 8d). At the peak of both episodes, temperature

and salinity at 21 m depth were nearly identical to those at 95 m depth, suggesting that a continuous, very weakly stratified layer extended through much of the water column.

Water properties during these episodes showed a clear tendency upward and rightwards in $\Theta$-$S_A$ space (Fig. 8e). This is in clear contrast to upper ocean warming periods elsewhere in the record (Fig. 7) and suggests that the observed warming in





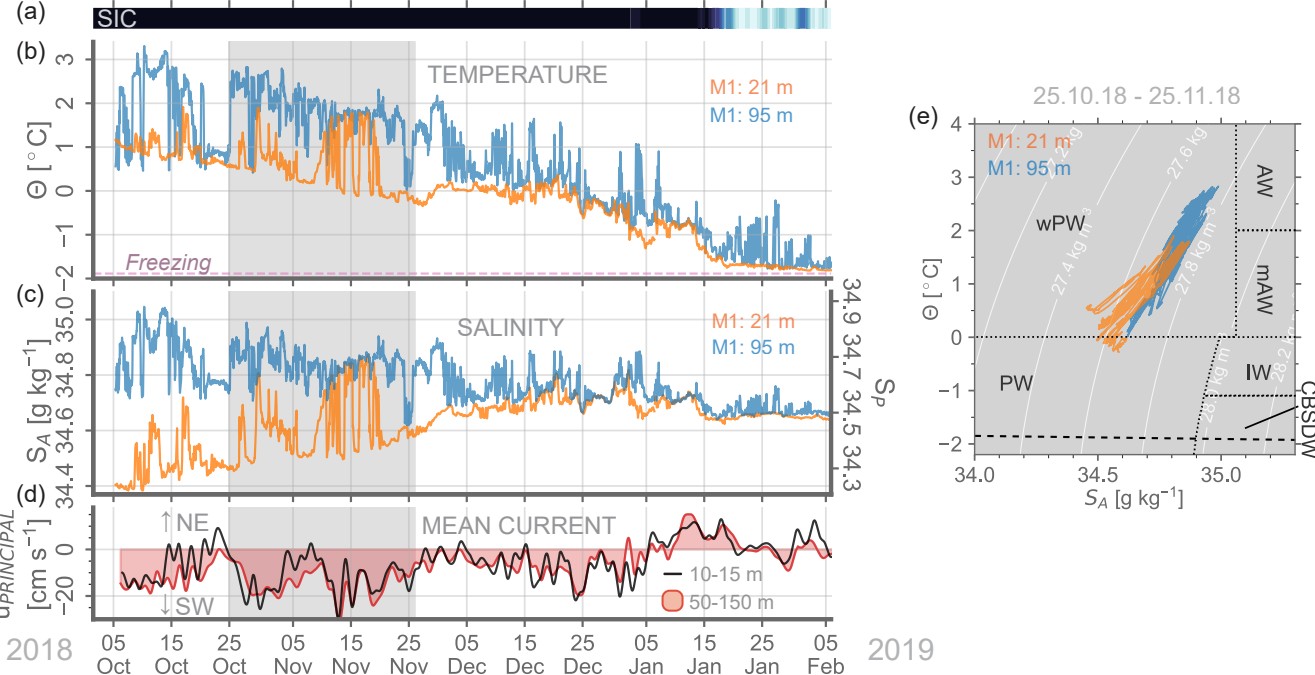

**Figure 8.** Upper ocean state at M1 during autumn 2018 to winter 2019. *a:* Local sea ice concentration (colour scale as in Fig. 3a, 4a). *b:* Hourly mean conservative temperature from the M1 mooring at 21 m (orange) and 95 m (blue) depth. Pink dashed line shows the freezing temperature at $S_A$ = 34.65 g kg$^{-1}$. *c:* Absolute salinity from M1 at 21 and 95 m depth. *d:* Mean subtidal currents at 10-15 m depth (black, from Sig500 instrument) and 50-150 m depth (red, from RDI150 instrument), both filtered as described in 2.1.4. *e:* Hourly data from the 21 m (orange) and 95 m (blue) sensors in $\Theta$-$S_A$ space. Only showing data from the period 25 October through 25 November, highlighted in grey in panels b-d.

autumn 2018 was due to AW influence rather than heat transfer from the atmosphere. Moreover, atmospheric temperatures
measured at the Kongsøya weather station indicate that the atmosphere was significantly colder than the upper ocean during
this period.

### 3.3 Shipboard CTD transects across flow pathways

CTD transects were conducted across Kvitøya Trough and in the cross-slope direction at the mooring sites within a week-
long period in late November 2019 (Fig. 9). All four transects show mid-depth temperature maxima which we interpret as
"cores" of AW in various stages of dilution—indicating flow into Kvitøya Trough from the north and cyclonic circulation of
AW in the nBS consistent with the mooring current measurements. November 2019 was not a period of particularly strong
mid-depth temperature maxima at either of the moorings (Fig. 3, 4): at M1, temperatures increased abruptly a few weeks
after the CTD transects were made, and at M2, a persistent mid-depth maximum had developed near 200 m, but daily mean
temperatures remained below -0.5°C. The 7-day separation between the first and last CTD profile means that the transects



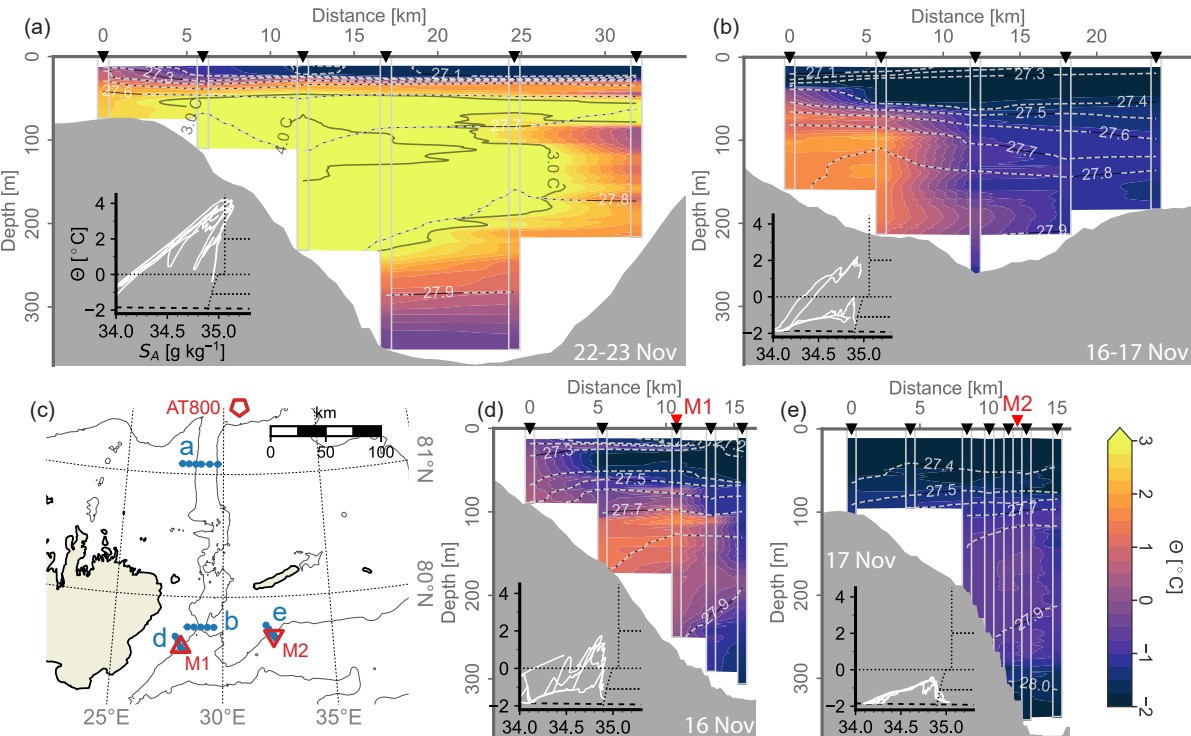

**Figure 9.** Shipboard CTD transects November 2019. Panels *a, b, d, e* show contours of Conservative Temperature (colour contours, gray contours above 3°C) and $\sigma_0$ (kg m$^{-3}$, labelled white/gray dashed line contours). Small inset plots show the transect CTD profiles in $\Theta, S_A$ space (white lines), see Fig. 6 for water mass categories indicated in dashed lines. Panel *c* shows the location of the transects (blue), plotted along with the 250 m contour (gray) and the mooring locations (red). Transect dates shown in each panel.

cannot be considered as a synoptic "snapshot" of the circulation; however, the transects in combination provide some insight into the distribution and pathways of AW-influenced water in the study area during a time of moderate or weak AW influence.

The northernmost transect (Fig. 9a) shows the southward intrusion of AW into Kvitøya Trough. A broad temperature maximum was found, centred in the western part of the trough near 100 m depth below a ∼25 m deep cold and fresh surface PW layer. In the warm core of the transect near 100 m, temperatures exceeded 4°C, and fell clearly within the $\Theta$-$S_A$ range of AW.

Further south in the trough, near M1, we observed a thicker (∼50 m) cold surface layer overlying a diluted AW core on the western side (Fig. 9b). The western AW layer extended to the bottom, but a core (∼ 2°C) was located near 100 m depth. Temperatures decreased rapidly towards the centre of the trough. On the eastern side of the transect, temperatures lay below 0°C throughout the water column.

Cross-slope transects at both mooring sites (Fig. 9de) show mid-depth temperature maxima with a warmer core centered

over the slope. Both transects also show isopycnals sloping downwards towards the northwest (towards shallower depth), consistent with southwestward geostrophic flow. At the M2 transect, temperatures in the core only reached ∼-0.4°C degrees,




**Table 3.** Correlation between $\Delta p_{K-B}$ and $\bar{U}_p$ (positive towards the $\sim$NE) shown in Fig. 10. Showing the maximum lagged correlation, with the time lag ($\Delta p_{K-B}$ leading $\bar{U}_p$) maximizing the correlation in parentheses. Calculated for the full available time series within 01.10.2018-01.10.2020, excluding "heavy sea ice periods", and only including "heavy sea ice" periods. Non-significant ($p <$0.05) correlation values shown in italics.

|  | M1 | M2 | AT800 |
| --- | --- | --- | --- |
| **Full period** | 0.47 (2d) | 0.27 (2d) | -0.18 (0d) |
| **W/o heavy sea ice** | 0.53 (2d) | 0.24 (4d) | -0.31 (0d) |
| **Heavy sea ice only** | 0.38 (2d) | 0.34 (2d) | *-0.14* (6d) |

falling entirely within the PW water mass range, but it was clearly distinguished from near-freezing temperatures above and below. At the M1 transect, core temperatures reached >1.7°C, with a $\Theta$-$S_A$ signature suggesting some mixing with AW, and with large fluctuations in the profiles indicating interleaving of water masses on the top of the core near 100 m depth.

### 3.4 Correlation between ocean currents and atmospheric pressure

Ocean currents displayed high subtidal variability on intraseasonal time scales (a few days to approximately one month) at both moorings, but particularly at M1 (Section 3.2). Especially in autumn, there were frequent episodes of abrupt intensification and slowdown of the currents, occasionally even reversal of the flow direction. These fluctuations also exerted a strong influence on the hydrography at shorter time scales, as shown by the covariability of currents and temperature in Fig. 3 and 8.

Suspecting that these fluctuations were atmospherically driven, we examined the relationship between currents and atmospheric pressure, which is also quite variable on sub-monthly time scales (Fig. 2b). 10 shows a comparison between mean subtidal mid-depth currents at the two moorings and the $\Delta p_{K-B}$ index (atmospheric pressure at Kvitøya minus atmospheric pressure at Bjørnøya, see Section 3.1). Through much of the study period, there is a clear relationship between $\Delta p_{K-B}$ and currents at M1, with negative $\Delta p_{K-B}$ associated with southwestward current anomalies and vice versa.

We evaluated the relationship quantitatively by computing correlations between daily time series of *i)* $\Delta p_{K-B}$ and *ii)* along-axis subtidal currents, positive towards the NE, depth-averaged between 125 m to 230 m (shorthand $\bar{U}_p$) after removing the 61-day running mean from both time series in order to focus on the relevant time scale. Correlations were evaluated for all three moorings with lags (atmosphere leading the ocean) spanning between 0 and 7 days and are shown in Table 3. The resulting maximum correlation between $\Delta p_{K-B}$ and $\bar{U}_p$ at M1 was $r = 0.47$ at a lag of two days, confirming a moderate positive correlation between the variables at M1. At M2, the maximum lagged correlation, also at a lag of two days, was positive but weak, and at AT800 it was negative but weak.

We recomputed correlations after removing parts of the record where sea ice concentration was consistently high in the nBS. Two time periods, Mar-Jun 2019 and Dec 2019-Mar 2020, were defined as *heavy sea ice periods* (Fig. 10ad). The correlation at M1 increased slightly when heavy sea ice periods were excluded and decreased when calculating the correlation *within* the



**Figure 10.** Unlagged time series of daily mean ocean currents, surface air pressure gradient, and sea ice concentration, after removing the 61-day running mean. *a, d:* nBS sea ice concentration (average in th A-nBS area, color scale as in Figures 3a, 4a). All remaining panels show $\Delta p_{K-B}$, the pressure differential Kvitøya-Bjørnøya (blue), and 125 m to 230 m depth averaged ocean currents in the principal direction (positive $\sim\rightarrow$NE) from ADCPs at different moorings (black/orange). *b, e:* Currents from M1, positive 45.2° CCW of east. *c, f:* Currents from M2, positive 35.7° CCW of east. *g:* Currents from AT800, positive 49.9° CCW of east (note that the y-axis scale has been reversed).

heavy sea ice periods (Table 3). At M2, the maximum correlations were higher during the heavy sea ice periods than in the remaining time series. At AT800, a significant correlation was not found when only including the heavy ice period.

Cross-spectral analysis of the time series of $\Delta p_{K-B}$ and $\bar{U}_p$ at M1 show significant coherence at a range of frequencies, with notable peaks in coherence squared near periods of 3.5, 11, and 18 days (Fig. A1). A lag of $\sim$2 days was typical across a wide range of frequencies (Fig. A1d), broadly consistent with the lagged correlation analysis.




**Figure 11.** Atmospheric variables in a regional subset of CARRA-EAST during the period 2018-2020, showing the average of daily mean conditions on days where $\Delta p_{K-B}$ exceeds 1.5 standard deviations above (upper) and below (lower) the mean. *a, g:* Average mean sea level pressure. *b, f:* Average wind at 10 m. The green contour indicates the magnitude of the average wind vectors. *c, g:* Average wind stress curl with average wind vectors. *d, h:* Like c, g, but zoomed in on a smaller region around Nordaustlandet-Kvitøya. Axis coordinates show the grid coordinates, which are equally spaced at 2.5 km resolution. Wind vectors (black) are subsampled every 15 th grid point in b, c, f, g and every 5th grid point in d, h.

We also evaluated the correlations between $\Delta p_{K-B}$ and $\bar{U}_p$ at M1, respectively, and the daily average wind stress curl computed from CARRA. Correlations were evaluated with time lags of ±7 days. The results indicate that both variables are weakly correlated with negative wind stress curl over the Kvitøya trough and over the FSAW core north of Svalbard (Fig. A2). Correlations with wind stress curl were generally maximized at a lag of 1-4 days for $\bar{U}_p$, and at zero lag for $\Delta p_{K-B}$. Looking more closely at the state of the atmosphere during negative and positive pressure meridional differences, Fig. 11 shows the regional patterns of sea level pressure, wind, and wind stress curl during periods in 2018-2020 where $\Delta p_{K-B}$ exceeded 1.5 standard deviations above or below its mean value (after subtracting the 61-day running mean). High $\Delta p_{K-B}$, corresponding to





lower pressure to the south and decreased ocean inflow at M1, was associated with easterly winds and negative wind stress curl in offshore areas north of Svalbard as well as in most of the Barents Sea. Conversely, periods of low $\Delta p_{K-B}$, corresponding to higher pressure to the south and stronger ocean inflow into the nBS, show a pattern of westerly winds and positive wind stress
curl in these same areas. During both phases, these patterns were often quite different close to the coasts as winds interact with the land mass. Notably, the mean wind stress curl during high $\Delta p_{K-B}$ changed sign from negative to positive toward the coast north of Svalbard (Fig. 11d). During low $\Delta p_{K-B}$, the study area encompassing M1 was found on the lee side of Nordaustlandet, in the vicinity of strong coastal wind stress anomalies of opposite sign to the north and south (Fig. 11h).

## 4    Discussion

### 4.1    Seasonal evolution of the water column

As might be expected, the records from the M1 and M2 moorings show that the upper ocean in the nBS is strongly influenced by the presence of sea ice. Strong freshening occurs during the melt season, and warming of the surface layer occurs during ice-free periods in summer, when heat can be transferred directly from the atmosphere to the surface ocean. Interestingly, the onset of freshening near 20 m depth occurred 1-2 months before the decrease in sea ice concentration, likely as a result of a
combination of local ice melting and thinning and advection of meltwater from regions nearer the ice edge. During both 2019 and 2020, surface freshening occurred earlier at M2 than at M1. This can be explained as a result of the progression of sea ice melting in the nBS, which during both seasons advanced from east to west.

The upper ocean preserves a lasting impact of the preceding sea ice season. Salinity at 20 m depth at M1 was 0.5 to 1.0 g kg$^{-1}$ higher in autumn 2018 (following a long ice-free summer) than in autumn 2019 (following a summer of persisting
ice cover in the western nBS). A crude estimate of freshwater content based on single-point sensors at M1 suggests that the freshwater fraction of the water column doubled between these periods, in rough agreement with Aaboe et al. (2021) who estimated a 1-m increase in freshwater content from 2018 to 2019. Interannual variations in wind forcing influences sea ice drift between the Arctic Ocean and the Barents Sea (Sorteberg and Kvingedal, 2006), and the high sea ice year of 2019 in the nBS was a result of large sea ice imports from the north and, to a lesser degree, east, during winter 2018/2019 (Aaboe et al.,
2021). The ice ultimately melted in the Barents Sea, supplying freshwater to the surface ocean during the melt season. Lind et al. (2018) showed that increased upper ocean freshening and strengthened salinity stratification as a result of strong sea ice imports reduces vertical mixing with deeper waters and preconditions the region for further sea ice growth the next winter. Based on our data, it is certainly conceivable that the absence of strong freshwater stratification in autumn 2018 was in part responsible for the late onset of the sea ice in the nBS this season (January 2019). However, the return of a thick and extensive
sea ice cover (Aaboe et al., 2021) and a fresh upper ocean layer in 2019 suggests that a sufficiently large import of sea ice can return the nBS to a cold, stratified, and ice covered state.

Interannual differences in salinity were not restricted to the upper ocean; at both moorings, the mid-depth waters were also fresher during spring 2020 than during spring 2019. While this could in part be related to varying rates of AW inflow during these two years, we mainly attribute it to different rates of meltwater being mixed down from the surface fresh layer, and




the sustained effect of meltwater inputs over one to several years. This is particularly the case at M1, where the 34.5 g kg$^{-1}$
isohaline frequently extended to ∼100 m depth during winter/spring 2020. The fresher water at mid-depth at M1 compared
to M2 may be explained by ocean advection through the Kvitøya-Nordaustlandet gap. While the nBS was largely ice covered
during early 2020, sea ice concentration north of Nordaustlandet and in the Kvitøya Trough area was more variable. In these
partially ice-free areas, mid-depth water masses influenced by increased wind-driven vertical mixing may have been advected

southward to M1 along with the prevailing ocean currents. This is supported by the relatively strong southwestward flow
observed at M1 (and largely absent at M2) during this period — including episodes of intensified southwestward currents
coinciding with intermittent cooling and freshening down to 175 m depth.

    The water masses observed at the moorings at mid-depth and below can generally be described as Polar Water mixed with
varying amounts of Atlantic Water. In addition to the freshwater influence described above, notable exceptions to this include *i)*

the relatively fresh and warm waters observed above 100 m M2 in Oct-Nov 2018, and *ii)* the cold, saline deep waters found near
250 m depth at M1 in spring 2019. We associate the first with downward mixing of fresher water masses that had been heated
at the surface in summer. The second we interpret as the influence of brine rejection and cooling during sea ice formation.
It is not clear whether these saline deep waters, which were close to surface freezing temperature, were produced locally or
transported from other areas. The high sea ice concentration at the M1 mooring location at the time may suggest the latter, as

there appears to have been few areas of open water locally where freezing could occur. This is in line with previous studies
showing that dense water is formed extensively on the banks and ends up filling the bottom of the deep trenches of the nBS
(Midttun, 1985; Årthun et al., 2011). However, it is possible that dense water formation occurred shoreward of M1, in small
coastal polynyas or other nearby shallow areas.

    The signature of Atlantic Water is recognizable as a mid-depth temperature maximum, varying in strength through the study

period. At M1, the maximum was typically found between 100 and 200 m depth during periods of strong AW influence, while
it was located below 150 m at M2. The vertical extent of the AW signal at M1 varied from spanning most of the water column
in autumn 2018, disappearing gradually in winter, reappearing as a more localised maximum around ∼150 depth in autumn
2019, and finally persisting as a weak warm layer below 150 m depth through most of 2020. The latter is the vertical structure
typically observed in the nBS (e.g., Loeng, 1991; Lind et al., 2016).

The AW layer observed at the mooring sites was generally situated below a colder and less dense layer extending to the
surface. This upper layer will have constituted a stratification barrier between the AW below and the ocean surface above,
reducing vertical mixing and preventing the transfer of heat from the AW layer and the upper ocean, sea ice, and lower
atmosphere. The near-surface waters varied in temperature depending on surface forcing, but they were distinguishable from
the AW layer by their lower salinity. However, an interesting exception was observed at the M1 mooring during autumn 2018,

when temperature and salinity near 20 m depth increased during episodes of strong inflow from the north, consistent with the
influence of AW in the uppermost part of the water column. Weak freshwater inputs during the unusually long 2018 ice-free
season may have resulted in a weak stratification which allowed AW to extend through the water column, and the upper ocean
temperature front to progress southward into the nBS. This finding is in line with Lind et al. (2018) in indicating that sea
ice melt plays a key role in regulating the vertical extent of the AW layer. Given the lack of freshwater inputs and relatively





vigorous wind forcing in autumn, it is likely that the AW influence extended to the ocean surface during these inflow periods. Since the air was substantially colder than the ocean, it is likely that the presence of warm AW-influenced waters at the surface resulted in increased transfer of heat from the ocean to the lower atmosphere. It may also have contributed to maintaining the ocean ice-free well into winter, until the massive sea ice inflows from the north began in January 2019 (Aaboe et al., 2021).

Compared to the AW-dominated ocean south of the Polar Front, the nBS can be described as a cold, stratified, PW-dominated
ocean environment (e.g., Loeng, 1991; Ingvaldsen et al., 2021). Our findings suggest that there are important local caveats to this general picture, as they show the strong influence of AW over large parts of the water column in inflow regions of the nBS during late autumn and winter. The observed seasonal hydrographic variability was to a large degree driven by inflows from the north, highlighting the strong influence that ocean advection excerts on the physical environment. The nBS experiences strong seasonality in general, and the seasonal timing of the AW inflow generally coincides with cooling of the atmosphere
and increase in sea ice concentration. Hydrographic surveys in the region have largely been based on shipboard measurements in late summer or early autumn — the mooring records presented in this study show the need for observations collected during different times of the year in order to accurately describe and monitor the ocean environment.

### 4.2 Circulation of Atlantic Water in the northern Barents Sea

The observed prevailing southwestward current direction at the two moorings is consistent with a topographically steered,
cyclonic ocean circulation in the northern Barents Sea. At both moorings, increases in southwestward flow coincided with increased mid-depth temperatures. This implies that some of the AW that is transported along with the FSAW north of Svalbard leaves the eastward path of the main current and enters the nBS at its northern boundary.

Cross-slope CTD transects collected in November 2019 indicate that the flow is concentrated over the bathymetric slope. This is typical of ocean circulation at high latitudes, where weak stratification and strong rotational influence combine to
vertically interlock the water column and steer flow along bathymetric contours. Surface forcing of the ocean over contours of constant potential vorticity favours cyclonic flow (Nøst et al., 2008), consistent with the predominant counterclockwise AW circulation in the Arctic Ocean and the Nordic Seas (Aagaard, 1989; Jones, 2001), and with the currents observed in the nBS in this study.

The southward transport of AW into troughs north of the Barents Sea has previously been indicated by model studies and
hydrographic measurements within the troughs (Aksenov et al., 2010; Pérez-Hernández et al., 2017; Menze et al., 2020). The two moorings in this study were positioned downstream of two troughs connecting the nBS with the FSAW, Kvitøya Trough (*KT*) and Franz-Victoria Through (*FVT*). The latter trough is far wider and deeper than the former, but our findings suggest that transport through KT has a profound influence on the hydrography in the study region north of Kong Karls Land.

We interpret our findings as follows: AW-influenced water masses enter the nBS through both the KT and the FVT, with the
core of the inflow concentrated over the topographic slope (Fig. 12). The KT constitutes the more direct route, and the AW arriving through this pathway tends to be warmer and less diluted. The water flowing through the broaderFVT and entering the study area from the east follows a longer trajectory after leaving the FSAW, and it is therefore subject to more surface influence and more mixing with Polar Water before arriving south of Kvitøya.




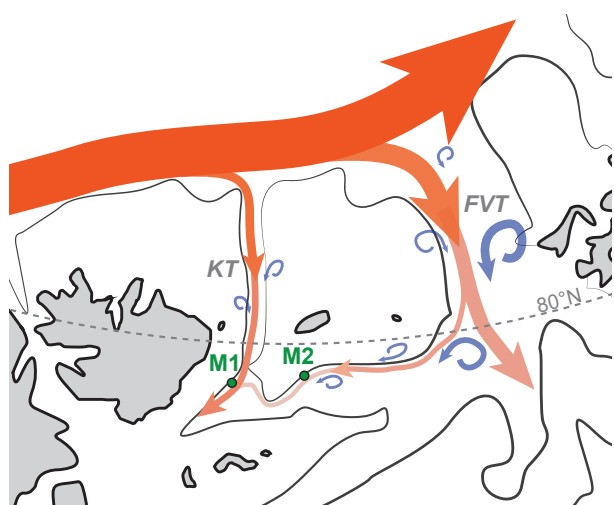

**Figure 12.** Conceptual interpretation of the advection of Atlantic Water into the study area during the inflow phase in late autumn/early winter. Orange arrows represent AW which is cooled and diluted along the pathways into the nBS. Blue arrows represent modification through lateral mixing with ambient water masses, and surface cooling and freshening. Black lines represent the ∼250 m depth contour.

The following observations from this study support this interpretation:

- The AW signal was more pronounced at M1 than at M2 during most of the study period; maximum temperatures were higher, and the vertical extent of the warm core greater, in line with it being closer to the source in the FSAW.

- Shipboard CTD transects from November 2019 suggest a continuous core of AW-influenced mid-depth waters extending from the northern opening of the KT to the M1 mooring. The signal was strongly diluted through the trough but remained clearly recognisable as a localised core at M1. An AW-influenced core was also present at M2 during the same period, but there it was significantly more diluted and somewhat deeper.

- AW arrived at the two mooring sites at different times. In particular, the seasonal inflow of AW at M1 seems to precede that at M2 by 1 to 2 months, consistent with changes in the FSAW taking longer to propagate through the FVT than the KT. The distance to M1 along the 250 m contour from the intersection of the FSAW with the opening of the KT is on the order of 200 km. From the same point following the contour eastward to the FVT, southward into the nBS and westward to M2, the distance is close to 450 km. Advection along the differential distance of 250 km over 1 to 2 months would imply a mean speed of 5 to 10 cm s$^{-1}$, a figure that seems reasonable given the observed currents.

- While AW-influenced water was frequently found above 100 m depth at M1, it was generally located below a colder, ∼150 m deep surface layer at M2. The longer pathway through FVT means that water masses arriving at M2 have been subject to more surface cooling and mixing with cold and fresh upper ocean water masses in the northeastern BS.





- Flow and water mass characteristics were generally more variable at M1. The inflow at M2 appeared more sluggish, with less frequent intermittent intensifications and reversals of the flow. This suggests that the flow through KT is relatively responsive to forcing. The flow along the longer pathway through the FVT, where the forcing is integrated over a longer time and a greater area, may effectively filter out upstream variability on shorter time scales by the time it arrives at M2.

It should be noted that since we do not have ocean observations to the east of M2, we cannot conclusively state that the
eastern pathway indicated in Fig. 12 is the origin of the warm water that arrived at this mooring. For example, some AW could also penetrate through the topographic depression extending southwestward on the eastern side of Kvitøya (Fig. 1), which forms a continuous pathway from M2 to the northern slope above ~170 m depth. However, since the AW maximum at M2 was typically located near 200 m depth, and the cross-slope CTD transect at M2 indicated an AW core situated deeper than the 200 m isobath, we consider it likely that the pathway indicated in Fig. 12 is the main one.

Heat transport in the FSAW undergoes a seasonal cycle associated with the eastward propagation of the seasonal cycle of AW from Fram Strait (Ivanov et al., 2009; Lique and Steele, 2012). The observed seasonal cycle in the hydrography at M1, with the arrival of AW in autumn, is consistent with previous mooring observations from the continental slope north of Svalbard (Renner et al., 2018; Lundesgaard et al., 2021), and supported at least qualitatively by the temperature evolution observed in the core of the FSAW north of Svalbard in 2019-2020. The seasonal cycle in the FSAW has a first-order impact of the on the
evolution of the water column at the M1 and M2 moorings. This underscores that advection from the north is a key driver of the ocean environment in the nBS.

The seasonal cycle in the FSAW affects the current strength as well as the water properties; the phase of maximum temperature is also associated with a maximum of eastward volume transport (Renner et al., 2018; Pérez-Hernández et al., 2019). This was also the case at M1 and M2 during the study period; the southwestward flow as well as the temperature peaked during late
autumn to early winter. It seems likely that the magnitude of the advection into the nBS is coupled to the strength of the FSAW over seasonal time scales, with increased eastward flow along the northern shelf slope resulting in increased transport through the troughs to the south. The dynamics governing the interactions between slope currents and submarine troughs are complex and may involve non-linear dynamics and instabilities of the along-slope flow (Dinniman and Klinck, 2004; Jordi et al., 2008; Nilsen et al., 2016). Since both the southward advection and the upstream heat content undergo seasonal cycles with similar
phase, both are likely to contribute to the seasonality in heat transport into the nBS.

Our findings document the entry of diluted AW into the nBS from the north, but they say little about the downstream pathways of the imported waters. The 250 m depth contour continues southwestward from the M1 mooring, but then turns sharply towards the east as the seafloor rises abruptly toward the small archipelago of Kong Karls Land. If the current core remains strongly constrained to constant depth, the flow might be expected to turn eastward and enter the eastern parts of the
nBS, where it might eventually mix with the ambient waters, or continue to circulate clockwise around Kong Karls Land. The bathymetric constraints are not absolute, however, and non-linear dynamics may come in to play, particularly near the sharp wedge in the bathymetry downstream of M1. Here, the small length scales may result in a large Rossby number, favouring ageostrophic flow (e.g., Phillips, 1963). Isobaths also diverge in this area, perhaps resulting in a slowing down of the currents and a widening of the current core. Nilsen et al. (2016) argued that at large Rossby number, the slope current core will tend



to deflect towards shallower depth in regions where the slope becomes less steep. Moreover, vorticity forcing associated with nearshore gradients in wind stress forcing may also act to bring some deeper water up onto the shelf. It is therefore not unlikely that some of the diluted AW observed at M1 may penetrate southward and onto the shelf west of Kong Karls Land.

The two-year record is too short to confidently distinguish between interannual and seasonal variations. However, it is interesting to observe that during 2018-2020, the onset of southwestward flow and mid-ocean warming occurred out of phase

with the warming of the atmosphere. As such, the area seems to experience successive maxima in incoming radiation (June), air temperature (mid-late summer), and warming of the ocean below the mixed layer (late autumn to early winter). A seasonal pulse of warm mid-ocean waters in autumn/winter may be a consistent feature of the ocean environment in the nBS. Occasionally, the seasonal pulse takes place below the sea ice cover, such as in 2019. The AW inflow branches frequently carry nutrients (Torres-Valdés et al., 2013) and marine organisms native to lower latitudes (Wassmann et al., 2015) into the Arctic – it is

unclear what role a seasonal AW pulse from the north into the dark and often sea ice covered nBS may play for the regional marine ecosystem.

On a note of caution, the mooring records shown in this study allow us to make qualified inferences about the circulation, but they are restricted each to a single location within inflow current cores. This study does not provide a comprehensive view of the regional circulation, nor does it attempt to quantify the advective fluxes of water, heat, and salt. At the time of writing,

additional ocean moorings are in place across the slope at M1 — these will hopefully help estimate the transports associated with the flow, and to study the dynamics of the slope current in more detail.

### 4.3 Atmospheric modulation of inflow from the north

Our findings demonstrate a correlation between the meridional atmospheric pressure gradient and the southwestward flow at depth at the M1 mooring (and to a lesser degree at M2). Low air pressure at Bjørnøya relative to Kvitøya, corresponding to

cyclonic, westward wind anomalies in the study area and over the FSAW, tend to coincide with a reduction of the southwestward ocean current, and vice versa. The high degree of flow variability appears to be in line with near-bottom currents measured in Olgastretet south and west of M1 by Sternberg et al. (2001). They reported relatively strong, predominantly southward near-bottom flows during autumn and winter, with large variability including flow reversals at periods of 3 to 8 days. The relationship with atmospheric forcing shown in this study appears qualitatively consistent with Aagaard et al. (1981), who

found that episodes of northward flow anomalies in Kvitøya Trough were associated with the passage of cyclones to the south.

Correlation analysis indicates that the atmospheric forcing leads the ocean response by approximately two days. The cause of this correlation is unclear, but we briefly discuss three possible mechanisms by which the meridional surface pressure gradient in the atmosphere could affect the ocean transport through Kvitøya Trough:

1. *Ocean response to zonal wind anomalies.* Since the atmospheric pressure gradient induces strong east-west winds over a
large area around Svalbard, we consider first the case of spatially uniform, zonal wind anomalies of changing sign. Wind-driven upwelling is typically a result of diverging Ekman transport (see Kämpf and Chapman, 2016), which may occur over the slope of a shallow shelf as a result of uniform along-shelf wind forcing. One could imagine that this mechanism





could bring additional Atlantic Water into the KT or even onto the shelf north of the nBS. However, as pointed out by Randelhoff and Sundfjord (2018), this type of shelf break upwelling is unlikely to occur north of Svalbard due to the

shelf break being relatively deep. Moreover, this mechanism would require easterly winds in order to generate shelf upwelling over a poleward deepening slope, and our observations find *decreased* inflow during easterly winds.

Another conceivable response to a uniform wind anomaly may involve the specifics of the coastal current interacting with Kvitøya Trough. Wind stress along the axis of a slope current can enhance cross-shelf flow through undersea canyons (Kämpf, 2006; She and Klinck, 2000), and such a mechanism has been observed in canyons elsewhere along the Arctic

Ocean continental slope (Carmack and Kulikov, 1998). However, the mechanism (summarized in Kämpf, 2006) requires upwelling-favourable winds to be directed opposite to the direction of trapped wave propagation — which, again, in this case would mean easterly winds. In summary, we find no obvious simple framework to explain the observed correlation as a result of uniform wind forcing. However, since the region in question is topographically and dynamically complex, these idealized conceptual frameworks may not be entirely appropriate for explaining the ocean response.

2. *Ocean response to wind stress curl anomalies.* Even in the absence of bathymetrical constraints, positive wind stress curl may set up divergent Ekman flows in the upper ocean, resulting in depression of the sea surface and upwelling from below. Conversely, negative wind stress curl will cause surface convergence, downwelling, and increased sea surface height. In a barotropic framework, a negative wind stress curl anomaly on the offshore side of a cyclonic slope current like the FSAW may therefore act to reduce the cross-slope pressure gradient and thereby weaken the current strength,

and vice versa. Kolås et al. (2020) found this mechanism to be at work further west in the FSAW, where a transition from negative to positive offshore wind stress curl resulted in a doubling of the current volume transport over a period of five days, and Renner et al. (2018) found upwelling and downwelling northeast of KT during strong wind stress curl episodes in December-January. Similarly, Nilsen et al. (2016) reported wind-driven increase of sea surface height on the shelf to result in an acceleration and shoreward shift of the WSC, resulting in increased flow of AW into troughs along

the West Spitsbergen coast.

We have shown that an atmospheric state favourable to the reduction or reversal in the southward ocean flow through KT is associated with easterly winds in the entire region around Svalbard, but also with negative wind stress curl over the FSAW as well as over KT (and vice versa). The wind stress curl anomaly is generally amplified on the offshore side of the FSAW. All else being equal, a negative/positive wind stress curl may be expected to produce offshore con-

vergence/divergence in the upper ocean, resulting in a weakening/strengthening of the offshore ocean sea surface height difference across the slope, and therefore also weakening/strengthening the barotropic component of a cyclonic slope current. The observed correlations therefore appear to be consistent with a mechanism where wind stress curl anomalies strengthen and weaken the FSAW and its branch running into KT, resulting in a modulation of the inflow into the nBS. However, the exact manner in which wind stress curl anomalies are reflected in ocean pressure gradients is more

complex; the actual wind stress curl pattern has a more complicated spatial structure than outlined above, and includes





gradients oriented *along* the flow path from the FSAW to the nBS through KT. Moreover, it is not clear exactly how a strengthening or weakening of the upstream slope current would translate to increased or reduced inflow at M1.

Lastly, it is not necessarily the case that the ocean response observed at the moorings is due to forcing far upstream. Wind anomalies are also experienced locally, and the wind stress curl patterns during strong forcing events show high amplitudes close to the coasts within the nBS as the winds blow around land topography. In particular, the coastal ocean on the leeward side of Nordaustlandet is exposed to strong wind stress curl forcing during westerly winds. Strong upper ocean convergence and divergence due to forcing acting locally in the nBS may induce coastal flow anomalies or eddies which could possibly diminish or amplify the background cyclonic circulation during inflow periods.

3. *Transient response to large upstream forcing events.* The weather systems in question are large-scale compared to the study region, and atmospheric pressure anomalies are likely to affect the FSAW-WSC current system over a large area on their way into the Barents Sea (Lien et al., 2013). Given the short time scales of the forcing and the ocean response, it is likely that the resulting ocean adjustment process involves the generation and propagation of topographically trapped waves (Mysak, 1980), which are commonly observed as a result of wind anomalies along shelf breaks in the high-latitude ocean (Inall et al., 2015; Spence et al., 2017). The complicated bathymetry means that the propagation of such waves might be rather complex in the nBS, but waves generated upstream in the FSAW or even in the WSC would likely propagate into KT and perhaps also the nBS. Here, coastal trapped waves might generate flow anomalies which may add to or subtract from the background circulation. The interaction of the FSAW with the KT could also be affected by remotely generated waves, as coastal trapped waves can affect up-and downwelling in submarine troughs (Saldías et al., 2021). In the present case, however, the short time lags suggest that a wave source of the anomalies observed at M1 would have to be located relatively close by, likely within the KT region.

In addition, the regional sea ice extent likely plays a role in the ocean response to large-scale atmospheric forcing, since sea ice modulates the momentum transfer from winds to the surface ocean. Increasing sea ice concentration from zero initially results in an increase in the transfer coefficient, but beyond a certain threshold (Martin et al., 2014, report $\sim$85%) it sharply decreases, and the energy transfer is inefficient at high sea ice concentrations where individual floes no longer move freely (Guest and Davidson, 1987; Cole et al., 2017). In the present study, we note that the decrease in ocean temperature coincided with the onset of >90% sea ice concentration north of Svalbard at the back end of both of the warmest periods at M1 (autumn 2018 and winter 2019). It is conceivable that rigid sea ice inhibiting wind forcing acted to shut down the southward ocean transport from this area, although we cannot confidently determine this based on our measurements.

Whatever the exact mechanism at work, it is clear that synoptic atmospheric pressure systems affect the flow into the nBS on sub-seasonal time scales. It is less clear whether they actually impact the net transport into the area. Nor is it obvious how the seasonal cycle in the advection of AW is related to wind forcing — the correlations described in the present study explicitly deal with time scales shorter than two months. Other studies have suggested that wind stress affects the strength and location of the Atlantic Water inflow north of Svalbard on time scales of weeks (Kolås et al., 2020), but also seasonally (Renner et al., 2018) and interannually (Lind et al., 2018).





A more complete understanding of the dynamics generating the observed circulation would require further investigation. In particular, the study of the connection between the nBS and the region to the north may be well-suited to regional numerical model experiments, idealised or otherwise. Such experiments may allow a more detailed study of the spatial structure of the ocean response and the explicit study of ocean momentum balance terms under varying forcing. Observational studies of the wind-driven ocean transport in this area would greatly benefit from multiple moorings located at different points along flow pathways.

*Data availability.* Data from the C3S Arctic Regional Regional Reanalysis (CARRA) were downloaded from the Copernicus Climate Change Service (C3S) Climate Data Store. AMSR2 sea ice concentration data (Spreen et al., 2008) were obtained from the University of Bremen Sea Ice Remote Sensing Data Centre (seaice.uni-bremen.de/start/data-archive). Weather station data were obtained from the climate database of the Norwegian Meteorological Institute (accessed through www.eklima.no). Shipboard CTD data will be made available through the Norwegian Marine Data Centre (nmdc.no) before the publication of this study. Mooring data will be made available at the Norwegian Polar Data Centre (data.npolar.no).

## Appendix A: Relationship between atmospheric forcing and ocean currents at M1 — additional figures

Figure A1 shows the coherence between the high-pass filtered meridional pressure difference, $\Delta p_{K-B}$, and the principal axis current at M1 averaged between 125 and 230 m depth, $\bar{U}_p$. We computed the auto- and cross-spectra of the daily average $\bar{U}_p$ and $\Delta p_{K-B}$ using the *spectra* module from *pycurrents* (available at https://currents.soest.hawaii.edu/hgstage/pycurrents). The $\bar{U}_p$ time series was linearly interpolated over a 15-day data gap from 02.11.2019 to 16.11.2019, and spectra were computed for the full study period between 06.10.2018 and 20.09.2020. Spectra were smoothed with a 9-point boxcar, and coherence was calculated from the smoothed spectra. The 95% confidence level for coherence squared was calculated as $1 - 0.95^{1/(EDOF-1)}$, with $EDOF = 9$, giving 0.31 (Fig. A1, black dashed line).

Figure A2 shows the spatial structure of the correlation between CARRA wind stress curl and $\Delta p_{K-B}$ and $\bar{U}_p$, respectively. The full available period was used for $\bar{U}_p$, and correlations with $\Delta p_{K-B}$ were restricted to the period 01.10.2018 - 01.10.2020. Correlations were computed at lags between -7 and +7 days.



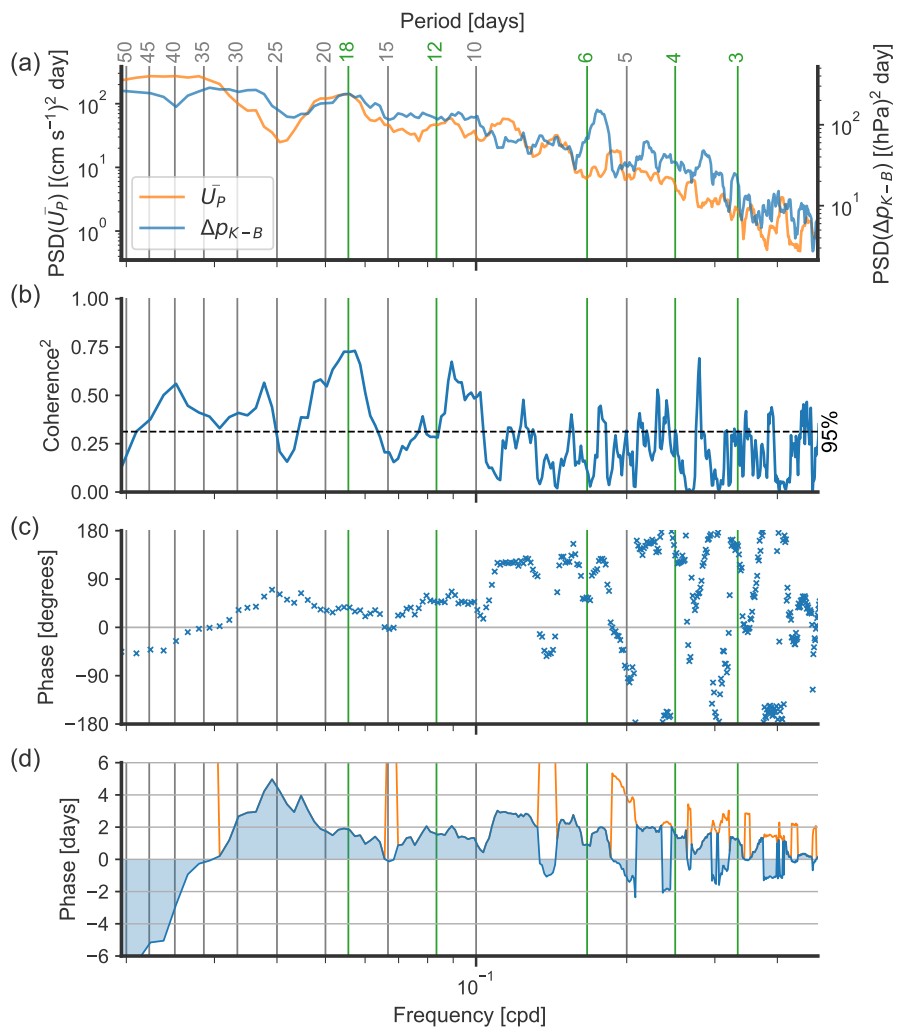

**Figure A1.** *a:* Power spectra of $\bar{U}_p$ and $\Delta p_{K-B}$. *b:* Coherence squared between the two variables, with the 95% confidence interval shown in the black dashed line. *c:* Coherence phase lag, with positive values indicating that $\Delta p_{K-B}$ leads $\bar{U}_p$. *d:* Coherence phase lag in units of days. The blue line shows the phase lag when the phase is allowed to be within ⟨-180°, 180°]. The orange line shows the lag in days when the phase is forced to be positive [0°, 360°⟩, corresponding to $\Delta p_{K-B}$ leading $\bar{U}_p$. In all panels, grey vertical lines indicate frequency values at equally spaced periods, and green vertical lines show frequencies of particular interest.

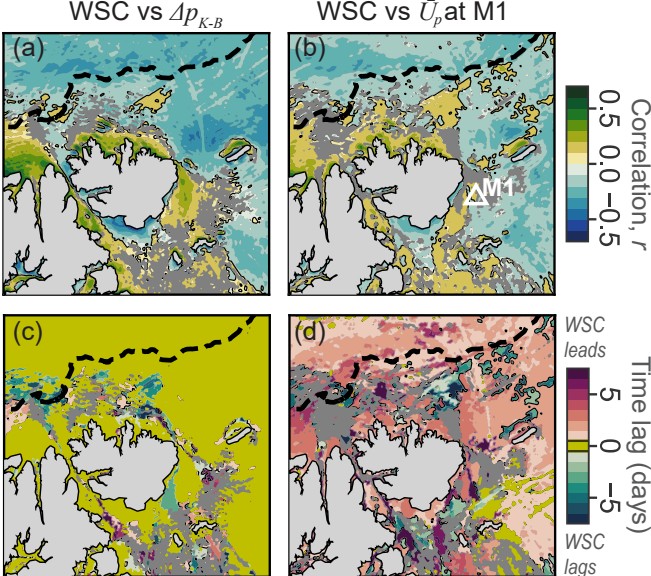

**Figure A2.** Correlation between CARRA reanalysis wind stress curl and *a)* $\Delta p_{K-B}$ (positive for low pressure anomaly at Bjørnøya), *b)* $\bar{U}_p$ at M1 (positive for flow towards the northeast). All variables have been high-pass filtered by subtracting the 61-day running mean. Correlations were computed for lags between -7 and +7 days. *c* and *d* show the lag at which maximum correlation shown in *a* and *b* is attained. Gray areas indicate where $p > 0.05$. The 800 m bathymethic contour shown in thick, dashed line.

*Author contributions.* Arild Sundfjord and Angelika H. H. Renner designed the study, collected the data, and contributed comments to the paper. Øyvind Lundesgaard performed the data processing and data analysis, produced the figures, and wrote the initial draft of the

830   manuscript. Sigrid Lind and Frank Nilsen contributed comments and discussion of importance to the subsequent development of the paper.

*Competing interests.* The authors declare that they have no conflict of interest.

*Acknowledgements.* The authors are grateful to NPI and IMR personnel, including captains and crews of the RV *Kronprins Haakon*, for technical support during the mooring cruises. The analysis was funded by the Research Council of Norway through the Nansen Legacy project (RCN 276730). Data collection was funded by the projects Nansen Legacy and A-TWAIN/SIOS-InfraNor (project number 66050).





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
