# Peer review of "Import of Atlantic Water and sea ice control the ocean environment in the northern Barents Sea"

_Ocean Science, 2022_

## Author Comment (AC2)

**Response to Reviewer #1**

We appreciate the helpful comments from Reviewer 1 on our submitted manuscript "Import of Atlantic Water and sea ice control the ocean environment in the northern Barents Sea". We believe that the constructive feedback has contributed to substantial improvements to the manuscript.

Below are our detailed responses to the review comments. Line numbers refer to the track-change manuscript.
* * *
As we understand the comments, the reviewer finds that the descriptive parts of the paper appear somewhat monotonous, and that interesting features of the data that are left without explanation. We understand the reviewer's concern; but we believe our approach is defensible. Given the nature of the dataset and the study, we have chosen to adhere relatively strictly to the IMRAD structure with an – admittedly – fairly dry description of the data in the Results section and the interpretation in the Discussion focused on a few specific topics. We believe that it is important to present the observational time series in their entirety as they provide important environmental context for other research. However, it is not our goal to explain all the features in the dataset; nor do we believe that we are necessarily able to do so. We have chosen instead to focus the discussion on a few key questions where we believe this study makes meaningful contributions to the state of knowledge. Based on the reviewer's comment, we have explained our approach more clearly at the beginning of the paper in order to better introduce the reader to the structure of the study (L109-110, L119-121).
* * *
The stronger freshening at M1 than at M2 is discussed in L619-625.
* * *
Both reviewers found major shortcomings in the section about atmosphere-ocean dynamical links. We acknowledge that this section had its weaknesses, and have tried to remedy them in the revised manuscript with good help from the suggestions from both reviewers. We have taken into account the suggestion from Referee #1 and performed correlation analyses between CARRA 10-m wind components the currents at both M1 and M2 (Fig. 10). We have made major changes to the text in section 3.4, and to some degree section 2.4 and 3.1, so as to better justify and motivate the use of the pressure difference between weather stations as an index of "atmospheric forcing". We believe that in the revised version, the pressure difference is better justified as a meaningful quantity (see in particular the paragraph at L540-554). We hope that the inclusion of a correlation analysis with CARRA winds, as well as the above modifications to the text, have addressed the reviewer's concern.
* * *
The reviewer makes a fair point about the estimate of freshwater content based on a very few points in the vertical: Lacking information about the vertical structure between the sensors, the linear assumption is perhaps too rough to be meaningful. In the reviewed manuscript, we have removed FWC altogether, and instead discuss our findings instead of salinity/temperature alone.
* * *
The reviewer requested an explanation of why we use Conservative Temperature (CT) in some cases and in-situ temperature (T) in others. This was done because we do not have CT available from all instruments; the RBR Solo instruments only measure T, not S, and we cannot therefore calculate CT. We acknowledge that this should have been made clearer to the reader, and have added an explanatory sentence in paragraph 1 of Section 3.2.1 (L328).
* * *
The reviewer reacted to the use of correlations between currents at different depths to support the idea that the observed currents were "vertically uniform". We acknowledge that the original phrasing of the sentence in question was misleading, and have reworded it in the revised version (L331-332). We have not removed the correlations from the text as we believe it provides useful information in combination with the figures.

---

## Author Comment (AC3)

**Response to Reviewer 2 (Yueng-Djern Lenn)**

We appreciate the insightful review of our paper by Dr. Lenn of our manuscript "Import of Atlantic Water and sea ice control the ocean environment in the northern Barents Sea". We found the comments very helpful, and we believe they have contributed to a better manuscript upon revision.

The reviewer found the part of the paper dealing with atmospheric forcing of inflow pulses less satisfactory. While we agree up to a point, and have addressed some of the specific concerns (see below), we would also like to stress that we cannot conclusively pin down the mechanism at work based on our data, and we want to remain completely open about this in the manuscript. What we can confidently say based on our observations is that there is a significant correlation between the ocean inflow and the synoptic atmospheric forcing. In the revised manuscript, we have stated more explicitly that we do not in fact believe that we can confidently determine the dynamical mechanisms at work based on our (single-point) current measurements.

Detailed comments are found below. Line numbers refer to the revised manuscript.
* * *
*Atmosphere-ocean mechanism:*

- Both reviewers reacted to the somewhat unmotivated use of the pressure differential between the two weather stations as an atmospheric forcing index used in the correlation analysis. In the revised manuscript, we have reorganized these parts of the paper, moving the introduction of the pressure difference index to the results section and spending more time justifying its use (L540-554). We have also included a correlation analysis between ocean currents and reanalysis winds (Fig. 10) in the preceding parts (L524-539), which hopefully makes the pressure index appear in a more logical sequence.

We have also taken up the reviewer's sensible suggestion of including a much larger region when showing correlations with atmospheric forcing (Figs. 10 and A2).

We have not included the appendix figures in the main manuscript. Instead, we hope that the inclusion of the new Figure 10 can serve to give the reader a good overview of the relationship between atmospheric forcing and ocean response over a larger spatial scale.

We have elaborated somewhat on remote vs local generation of the observed signals in Section 4.3 (L838-845). We have not made an attempt to estimate wave speeds directly (a rather complicated exercise involving detailed knowledge of wave dynamics, scales, stratification, and topography), but have instead made reference to the phase speeds reported by Inall et al 2015 in western Spitsbergen. We have also modified the statement in question to not so categorically discount remotely generated waves.

*Minor points:*

- We appreciate the suggestion to broaden the references in the Introduction. The revised version includes a wider and more comprehensive set of references, including relevant papers by Schauer and Mauritzen. We have not removed the previous references as we find that they also provide meaningful background context to this study (the knowledge base has expanded significantly during the last 10-20 years).

- We have followed the reviewer's suggestion of writing out certain acronyms in order to facilitate easier reading and avoid confusion with water masses. "nBS" has been replaced with "northern Barents Sea", "KT" with "Kvitøya Trough", etc. We have left abbreviations of water masses and named ocean currents in line with standard practice.

- We appreciate the suggestion to clarify the language in order to make a clearer distinction between implied advection and local processes. In the revised manuscript, we have avoided words like "warming" and "freshening" unless local processes, as opposed to advection, are implied.

- We take the reviewer's point in that the discussion about freshwater and sea ice could be strengthened by a more quantitative analysis. Unfortunately, we believe that our options are somewhat limited given that we have few sensors to work with (one CTD sensor near 20 m). Given the major advective sea ice imports, it is also difficult to distinguish between sea ice increase due to local freezing versus advection from other regions. We agree with the reviewer that a more quantitative assessment of the ice-air-mixed layer interplay would strengthen this part of the paper, but have avoided estimates requiring extrapolating from the data beyond what we believe is reasonable. For example, any estimate of the depth of the mixed layer, the structure of the haline and thermal stratification, or integrated heat/salinity budget terms for the mixed layer would be of great interest, but unfortunately, we cannot estimate any of these based on our data. (As pointed out by Reviewer #1, exactly the same can be said about estimating freshwater content based on very few sensors in the vertical. We have therefore elected to remove the freshwater content estimate altogether in the updated manuscript, sticking to a discussion in terms of salinity.) We have tried to make our approach clear in the Discussion (L611-614).